# UNNORMALIZED DENSITY ESTIMATION WITH ROOT SOBOLEV NORM REGULARIZATION

## ABSTRACT

We propose a new approach to non-parametric density estimation that is based on regularizing a Sobolev norm of the density. This method is statistically consistent, is different from Kernel Density Estimation, and makes the inductive bias of the model clear and interpretable. While there is no closed analytic form for the associated kernel, we show that one can approximate it using sampling. The optimization problem needed to determine the density is non-convex, and standard gradient methods do not perform well. However, we show that with an appropriate initialization and using natural gradients, one can obtain well performing solutions. Finally, while the approach provides unnormalized densities, which prevents the use of log-likelihood for cross validation, we show that one can instead adapt Fisher Divergence based Score Matching methods for this task. We evaluate the resulting method on the comprehensive recent Anomaly Detection benchmark suite, ADBench, and find that it ranks second best, among more than 15 algorithms.

## 1 INTRODUCTION

Density estimation is one of the central problems in statistical learning. In recent years, there has been a tremendous amount of work in the development of parametric neural network based density estimation methods, such as Normalizing Flows Papamakarios et al. (2021), Neural ODEs Chen et al. (2018), and Score Based methods, Song et al. (2021). However, the situation appears to be different for non parametric density estimation methods, Wasserman (2006), Härdle et al. (2004). While there is recent work for low dimensional (one or two dimensional) data, see for instance Takada (2008), Uppal et al. (2019), Cui et al. (2020), Ferraccioli et al. (2021) (see also the survey Kirkby et al. (2023)), there still are very few non-parametric methods applicable in higher dimensions.

Compared to parametric models, non parametric methods are often conceptually simpler, and the model bias (e.g., prior knowledge, type of smoothness) is explicit. This may allow better interpretability, and better regularization control in smaller data regimes.

Let $\mathcal{S} = \{x_i\}_{i=1}^N \subset \mathbb{R}^d$ be a set of data points sampled i.i.d from some unknown distribution. In this paper we introduce and study a density estimator of the following form:

$$f^* := \underset{f \in \mathcal{H}^a}{\operatorname{argmin}} -\frac{1}{N} \sum_{i=1}^N \log f^2(x_i) + \|f\|_{\mathcal{H}^a}^2 . \tag{1}$$

Here $\mathcal{H}^a$ is a Sobolev type Reproducing Kernel Hilbert Space (RKHS) of functions, having a norm of the form

$$\|f\|_{\mathcal{H}^a}^2 = \int_{\mathbb{R}^d} f^2(x)dx + a \int_{\mathbb{R}^d} |(Df)|^2 (x)dx, \tag{2}$$

where $D$ represents a combination of derivatives of a certain order. Importantly, the order of these derivatives is required to exceed $d/2$ (see Therom 2). The density estimate is given by the function $(f^*)^2$. Note that $(f^*)^2$ is clearly non-negative, and $\|f\|_{\mathcal{H}^a} < \infty$ implies $\int_{\mathbb{R}^d} (f^*)^2(x)dx < \infty$. Thus $(f^*)^2$ is integrable, although not necessarily integrates to 1. Note also that (1) is essentially a regularized maximum likelihood estimate, where in addition to bounding the total mass of $(f^*)^2$, we also bound the norm of the derivatives of $f^*$ of certain order. The fact that $\mathcal{H}^a$ is an RKHS allows us to compute $f^*$ via the standard Representer Theorem. Observe that it would not be possible to

control only the norm $L_2$ norm of $f^*$ and maintain computabilty, since $L_2$ is not an RKHS. However, adding the derivatives with any coefficient $a > 0$ makes the space into an RKHS which allows to control smoothness, hence, we call the objective Root Sobolev Regularized density estimator (RSR). Notice that this also provides implicit control over $||f^*||_{L_2}$, a desirable but relatively uncommon attribute of unnormalized density estimators.

Despite it being natural and simple, the objective (1) has not been studied in the literature as a tool for multidimensional data analysis. It has been introduced in Good and Gaskins (1971) and further studied in Klonias (1984), in the context of spline based methods in *one* dimension. Our goal in this paper is to develop the necessary ingredients for RSR to become useful in high dimensions. Specifically, for $d > 1$, the kernel corresponding to $\mathcal{H}^a$, which we call the SDO kernel (Single Derivative Order; see Section 4), no longer has an analytical expression. However, we show that nevertheless, it can be approximated by an appropriate sampling procedure. Next, standard gradient descent optimization of (1) produces poor results. We show that this may be improved by an appropriate initialization, and further improved by using a certain *natural gradient* rather than the standard one.

As mentioned before, the solutions of (1) are unnormalized (see also the discussion below). This introduces a particular nuance in the context of hyperparameter tuning, as it prevents the utilization of the maximum likelihood measure to establish the optimal parameter. To bypass these normalization challenges, we apply a score-based method for measuring the Fisher Divergence (FD), which uses log-likelihood *gradients* for divergence measurement, thereby eliminating the need for normalization. More specifically, we incorporate the concept of score-matching (Hyvärinen and Dayan, 2005; Song et al., 2020; Song and Ermon, 2019), a technique that has recently garnered renewed interest.

In addition to the above contributions, we provide a family of examples where one prove that RSR and the standard Kernel Density Estimator (with the same kernel) may arbitrarily differ. Thus, RSR is a genuinely new estimator, with different properties than Kernel Density Estimation (KDE). We also show that examples as above occur naturally in real datasets. Finally, we prove the consistency of the RSR estimator for the SDO kernels in any fixed dimension $d$, under mild assumptions on the ground truth density generating $x_i$'s.

Note that as a result of consistency, it can also be shown that the estimator $(f^*)^2$ becomes normalized, at least asymptotically with $N$. However, for finite $N$, computing or even estimating the normalization constant is not straightforward, and is outside the scope of this paper. Instead, we will focus on the Anomaly Detection (AD) applications, that do not require a normalization. Indeed, AD is based on the comparison of the likelihoods of different points, requiring only the ratios of the density values at these points. Note also that standard MCMC sampling algorithms, such as Langevin Dynamics or Hamiltonian Monte Carlo, do not require the knowledge of the normalization.

With this collection of contributions in place, we show that RSR achieves the remarkable performance of scoring *second best* on a recent comprehensive Anomaly Detection benchmark, Han et al. (2022), which includes more than 45 datasets and more than 15 specialized AD methods.

The rest of the paper is organized as follows: Section 2 reviews related literature. Section 3 introduces the RSR estimator and treats associated optimization questions. The SDO kernel and the associated sampling approximation are discussed in Section 4. Section 5 provides an example where RSR differs from KDE. Section 6 contains the experimental results. Consistency is discussed in Section 2, and the full proof, along with a proof overview, can be found in Supplementary Material Section L.

## 2 LITERATURE AND RELATED WORK

A as discussed in Section 1, a scheme that is equivalent to (1) was studied in Good and Gaskins (1971) and Klonias (1984); see also Eggermont et al. (2001). However, these works concentrated solely on 1d case, and used spline methods to solve (3) in the special case that amounts to the use of one particular kernel. Our more general RKHS formulation in Section (3.1) allows the use of a variety of kernels. Most importantly, however, as discussed in Section (1), in this work we have developed and evaluated the high dimensional version of RSR.

The most common non parametric density estimator is the Kernel Density Estimator (KDE), Härdle et al. (2004); Wasserman (2006). For comparison, we have evaluated KDE, with the two most popular kernels, Gaussian and Laplacian, on the AD benchmark. However, these methods did not perform

well (Section 6.1) on this task. We have also evaluated KDE with the SDO kernel that we introduce in Section 4, and which has not been previously considered in the literature for $d > 1$. Remarkably, we find that using this kernel significantly improves the AD performance compared to Gaussain and Laplacian kernels. However, the performance is still subpar to the RSR estimator.

Another common group of non parametric density estimators are the *projection methods*, Wainwright (2019). These methods have mostly been studied in one dimensional setting, see the survey Kirkby et al. (2023). It is worth noting that with the exception of Uppal et al. (2019), the estimators produced by these methods are not densities, in the sense that they do not integrate to 1, but more importantly, may take negative values. In the context of minmax bounds, projection methods in high dimensions were recently analyzed in Singh et al. (2018), extending a classical work Kerkyacharian and Picard (1993). However, to the best of our knowledge, such methods have never have been practically applied in high dimensions.

Fisher Divergence (FD) is a similarity measure between distributions, which is based on the score function – the gradient of the log likelihood. In particular, it does not require the normalization of the density. The divergence between data and a model can be approximated via the methods of Hyvärinen and Dayan (2005), which have been recently computationally improved in Song et al. (2020) in the context of score based generative models, Song and Ermon (2019). Here we use the FD as a quality metric for hyperparameter selection. In particular, we adapt the Hutchinson trace representation based methods used in Song et al. (2020) and Grathwohl et al. (2018) to the case of models of the form (3). Full details are given in the Supplementary Material Section H.

The concept of a gradient that is independent of a parametrisation was proposed in Amari (1998), and in Mason et al. (1999). In Kakade (2001) it was introduced into Reinforcement Learning, where it is widely used today. Here we consider specifically a Hilbert Space version of the notion, which also has a variety of applications, although typically not in RL. See for instance Mason et al. (1999), Yao et al. (2007), and Shen et al. (2020) for a sample of early and more recent applications. Natural Gradient in Hilbert Spaces is also referred to as Functional Gradient in the literature. While we are not aware of a dedicated treatment of the subject, introductory notes may be found at Bagnell (2012) and in the works cited above.

Consistency of the RSR in one dimension was shown in Klonias (1984), for kernels that coincide with SDO in one dimension. In Supplementary Material Section L we state and prove the $L_2$ consistency of the RSR in any fixed dimension $d$, for compactly supported ground truth densities. Our approach generally follows the same lines as that of Klonias (1984). However, some of the estimates are done differently, since the corresponding arguments in Klonias (1984) were intrinsically one dimensional.

## 3 THE RSR DESNITY ESTIMATOR

In this Section we describe the general RSR Density Estimation Framework, formulated in an abstract Reproducing Kernel Hilbert Space. We first introduce the general optimization problem and discuss a few of its properties. In Section 3.2 we discuss the gradient descent optimization and introduce the natural gradient.

### 3.1 THE BASIC FRAMEWORK

Let $\mathcal{X}$ be a set and let $\mathcal{H}$ be a Reproducing Kernel Hilbert Space (RKHS) of functions on $\mathcal{X}$, with kernel $k : \mathcal{X} \times \mathcal{X} \to \mathbb{R}$. In particular, $\mathcal{H}$ is equipped with an inner product $\langle \cdot, \cdot \rangle_{\mathcal{H}}$ and for every $x \in \mathcal{X}$, the function $k(x, \cdot) = k_x(\cdot) : \mathcal{X} \to \mathbb{R}$ is in $\mathcal{H}$ and satisfies the reproducing property, $\langle k_x, f \rangle_{\mathcal{H}} = f(x)$ for all $f \in \mathcal{H}$. The norm on $\mathcal{H}$ is denoted by $\|f\|_{\mathcal{H}}^2 = \langle f, f \rangle_{\mathcal{H}}$, and the subscript $\mathcal{H}$ may be dropped when it is clear from context. We refer to Schölkopf et al. (2002) for a general introduction to RKHS theory.

Given a set of points $S = \{x_1, \ldots, x_N\} \subset \mathcal{X}$, we define the RSR estimator as the solution to the following optimization problem:

$$f^* = \operatorname*{argmin}_{f \in \mathcal{H}} -\frac{1}{N} \sum_i \log f^2(x_i) + \|f\|_{\mathcal{H}}^2 . \tag{3}$$

As discussed in Section 1, for appropriate spaces $\mathcal{H}$, the function $(f^*)^2$ corresponds to an unnormalized density (That is, $\int_{\mathbb{R}^d} (f^*)^2(x)dx < \infty$, but not necessarily $\int_{\mathbb{R}^d} (f^*)^2(x)dx = 1$). We now discuss a few basic properties of the solution to (3). First, by the Representer Theorem for RKHS, the minimizer of (3) has the form

$$f(x) = f_\alpha(x) = \sum_{i=1}^N \alpha_i k_{x_i}(x), \text{ for some } \alpha = (\alpha_1, \ldots, \alpha_N) \in \mathbb{R}^N. \tag{4}$$

Thus one can solve (3) by optimizing over a finite dimensional vector $\alpha$. Next, it is worth noting that standard RKHS problems, such as regression, typically use the term $\lambda \|h\|_{\mathcal{H}}^2$, where $\lambda > 0$ controls the regularization strength. However, due to the special structure of (3), any solution with $\lambda \neq 1$ is a rescaling by a constant of a $\lambda = 1$ solution. Thus considering only $\lambda = 1$ in (3) is sufficient. In addition, we note that any solution of (3) satisfies $\|f\|_{\mathcal{H}}^2 = 1$. See Lemma 4 in Supplementary Material for full details on these two points.

Next, observe that the objective

$$L(f) = -\frac{1}{N} \sum_i \log f^2(x_i) + \|f\|_{\mathcal{H}}^2 = -\frac{1}{N} \sum_i \log \langle f, k_{x_i} \rangle_{\mathcal{H}}^2 + \|f\|_{\mathcal{H}}^2 \tag{5}$$

is not convex in $f$. This is due to the fact that the scalar function $a \mapsto -\log a^2$ from $\mathbb{R}$ to $\mathbb{R}$ is not convex and is undefined at 0. However, the restriction of $a \mapsto -\log a^2$ to $a \in (0, \infty)$ is convex. Similarly, the restriction of $L$ the positive cone of functions $\mathcal{C} = \{f \mid f(x) \geq 0 \ \forall x \in \mathcal{X}\}$ is convex. Empirically, we have found that the lack of convexity results in poor solutions found by gradient descent. Intuitively, this is caused by $f$ changing sign, which implies that $f$ should pass through zero at some points. If these points happen to be near the test set, this results in low likelihoods. At the same time, there seems to be no computationally affordable way to restrict the optimization to the positive cone $\mathcal{C}$. We resolve this issue in two steps: First, we use a non-negative $\alpha$ initialization, $\alpha_i \geq 0$. Note that for $f$ given by (4), if the kernel is non-negative, then $f$ is non-negative. Although some kernels are non-negative, the SDO kernel, and especially its finite sample approximation (Section 4.2) may have negative values. At the same time, there are few such values, and empirically such initialization tremendously improves the performance of the gradient descent. Second, we use the *natural gradient*, as discussed in the next section. One can show that for non-negative kernels, $\mathcal{C}$ is in fact invariant under natural gradient steps (supplementary material Section G). This does not seem to be true for the regular gradient. Empirically, this results in a more stable algorithm and further performance improvement. A comparison of standard and natural gradients w.r.t negative values is given in Section 6.3.

## 3.2 Gradients and Minimization

We are interested in the minimization of $L(f)$, defined by (5). Using the representation (4) for $x \in \mathcal{X}$, we can equivalently consider minimization in $\alpha \in \mathbb{R}^N$. Let $K = \{k(x_i, x_j)\}_{i,j \leq N} \in \mathbb{R}^{N \times N}$ denote the empirical kernel matrix. Then standard computations show that $\|f_\alpha\|_{\mathcal{H}}^2 = \langle K\alpha, \alpha \rangle_{\mathbb{R}^N}$ and we have $(f_\alpha(x_1), \ldots, f_\alpha(x_N)) = K\alpha$ (as column vectors). Thus one can consider $L(f_\alpha) = L(\alpha)$ as a functional $L : \mathbb{R}^N \to \mathbb{R}$ and explicitly compute the gradient w.r.t $\alpha$. This gradient is given in (6).

As detailed in 3.1, it is also useful to consider the Natural Gradient – the gradient of $L(f)$ as a function of $f$, directly in the space $\mathcal{H}$. Briefly, a directional Fréchet derivative, Munkres (2018), of $L$ at point $f \in \mathcal{H}$ in direction $h \in \mathcal{H}$ is defined as the limit $D_h L(f) = \lim_{\varepsilon \to 0} \varepsilon^{-1} \cdot (L(f + \varepsilon h) - L(f))$. As a function of $h$, $D_h L(f)$ can be shown to be a bounded and linear functional, and thus by the Riesz Representation Theorem, there is a vector, which we denote $\nabla_f L$, such that $D_h L(f) = \langle \nabla_f L, h \rangle$ for all $h \in \mathcal{H}$. We call $\nabla_f L$ the Natural Gradient of $L$, since its uses the native space $\mathcal{H}$. Intuitively, this definition parallels the regular gradient definition, but uses the $\mathcal{H}$ inner product to define the vector $\nabla_f L$, instead of the standard, "parametrization dependent" inner product in $\mathbb{R}^N$, that is used to define $\nabla_\alpha L$. For the purposes of this paper, it is sufficient to note that similarly to the regular gradient, the natural gradient satisfies the chain rule, and we have $\nabla_f \|f\|_{\mathcal{H}}^2 = 2f$ and $\nabla_f \langle g, f \rangle_{\mathcal{H}} = g$ for all $g \in \mathcal{H}$. The explicit gradient expressions are given below:

**Lemma 1** (Gradients). *The standard and the natural gradients of $L(f)$ are given by*

$$\nabla_\alpha L = 2 \left[ K\alpha - \frac{1}{N} K(K\alpha)^{-1} \right] \in \mathbb{R}^N \text{ and } \nabla_f L = 2 \left[ f - \frac{1}{N} \sum_{i=1}^N f^{-1}(x_i) k_{x_i} \right] \in \mathcal{H} \tag{6}$$

*where for a vector $v \in \mathbb{R}^d$, $v^{-1}$ means coordinatewise inversion.*

If one chooses the functions $k_{x_i}$ as a basis for the space $H_S = span\{k_{x_i}\}_{i \leq N} \subset \mathcal{H}$, then $\alpha$ in (4) may be regarded as coefficients in this basis. For $f = f_\alpha \in H_S$ one can then write in this basis $\nabla_f L = 2\left[\alpha - \frac{1}{N}(K\alpha)^{-1}\right] \in \mathbb{R}^N$. Therefore in the $\alpha$-basis we have the following standard and natural gradient iterations, respectively:

$$\alpha \leftarrow \alpha - 2\lambda\left[K\alpha - \frac{1}{N}K(K\alpha)^{-1}\right] \text{ and } \alpha \leftarrow \alpha - 2\lambda\left[\alpha - \frac{1}{N}(K\alpha)^{-1}\right], \tag{7}$$

where $\lambda$ is the learning rate.

## 4 SINGLE DERIVATIVE ORDER KERNEL APPROXIMATION

In this Section we introduce the Single Derivative Order kernel, which corresponds to norms of the form (2) discussed in Section 1. In Section 4.1 we introduce the relevant Sobolev functional spaces and derive the Fourier transform of the norm. In Section 4.2 we describe a sampling procedure that can be used to approximate the SDO.

### 4.1 THE KERNEL IN INTEGRAL FORM

For a function $f : \mathbb{R}^d \to \mathbb{C}$ and a tuple $\kappa \in (\mathbb{N} \cup \{0\})^d$, let $D^\kappa = \frac{\partial f}{\partial x_1^{\kappa_1} \dots \partial x_d^{\kappa_d}}$ denote the $\kappa$ indexed derivative. By convention, for $\kappa = (0, 0, \dots, 0)$ we set $D^\kappa f = f$. Set also $\kappa! = \prod_{j=1}^d \kappa_j!$ and $|\kappa|_1 = \sum_{j=1}^d \kappa_j$. Set $\|f\|_{L_2}^2 = \int |f(x)|^2 dx$. Then, for $m \in \mathbb{N}$ and $a > 0$ denote

$$\|f\|_a^2 = \|f\|_{L_2}^2 + a \sum_{|\kappa|_1 = m} \frac{m!}{\kappa!} \|(D^\kappa f)\|_{L_2}^2. \tag{8}$$

The norm $\|f\|_a^2$ induces a topology that is equivalent to that of a standard $L_2$ Sobolev space of order $m$. We refer to Adams and Fournier (2003), Saitoh and Sawano (2016) for background on Sobolev spaces. However, here we are interested in properties of the norm that are finer than the above equivalence. For instance, note that for all $a \neq 0$ the norms $\|f\|_a$ are mutually equivalent, but nevertheless, a specific value of $a$ is crucial in applications, for regularization purposes.

Let $\mathcal{H}^a = \left\{f : \mathbb{R}^d \to \mathbb{C} \mid \|f\|_a^2 < \infty\right\}$ be the space of functions with a finite $\|f\|_a^2$ norm. Denote by

$$\langle f, g\rangle_{\mathcal{H}^a} = \langle f, g\rangle_{L_2} + a \sum_{|\kappa|_1 = m} \frac{m!}{\kappa!} \langle(D^\kappa f), (D^\kappa g)\rangle_{L_2} \tag{9}$$

the inner product that induces the norm $\|f\|_a^2$.

**Theorem 2.** *For $m > d/2$ and any $a > 0$, the space $\mathcal{H}^a$ admits a reproducing kernel $k^a(x, y)$ satisfying $\langle k_x^a, f\rangle_{\mathcal{H}^a} = f(x)$ for all $f \in \mathcal{H}^a$ and $x \in \mathbb{R}^d$. The kernel is given by*

$$k^a(x, y) = \int_{\mathbb{R}^d} \frac{e^{2\pi i\langle y-x, z\rangle}}{1 + a \cdot (2\pi)^{2m} \|z\|^{2m}} dz = \int_{\mathbb{R}^d} \frac{1}{1 + a \cdot (2\pi)^{2m} \|z\|^{2m}} \cdot e^{2\pi i\langle y, z\rangle} \cdot \overline{e^{2\pi i\langle x, z\rangle}} dz. \tag{10}$$

The proof of Theorem 2 follows the standard approach of deriving kernels in Sobolev spaces, via computation and inversion of the Fourier transform, see Saitoh and Sawano (2016). However, compact expressions such as (10) are only possible for some choices of derivative coefficients. Since the particular form (8) was not previously considered in the literature (except for $d = 1$, see below), we provide the full proof in the Supplementary Material.

### 4.2 KERNEL EVALUATION VIA SAMPLING

To solve the optimization problem (3) in $\mathcal{H}^a$, we need to be able to evaluate the kernel $k^a$ at various points. For $d = 1$, closed analytic expressions were obtained in cases $m = 1, 2, 3$ in Thomas-Agnan

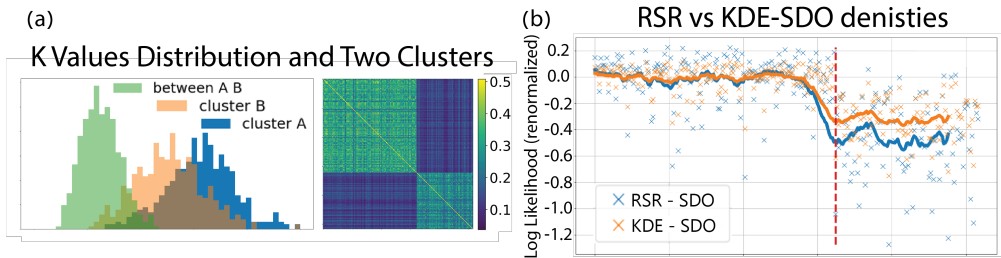

Figure 1: (a) Distribution of Kernel Values and SDO Kernel Values Inside and Between Clusters. (b) RSR and KDE Loglikelihoods. the x-axis represents points in the data, arranged by clusters, y-axis shows the log-likelihood.

(1996). In particular, for $m = 1$, $k^a$ coincides with the Laplacian kernel $k_h(x, y) = e^{-h|x-y|}$. However, for $d > 1$, it seems unlikely that there are closed expressions. See Novak et al. (2018) for a discussion of this issue for a similar family of norms.

To resolve this, note that the form (10) may be interpreted as an average of the terms $e^{2\pi i \langle y, z \rangle} \cdot \overline{e^{2\pi i \langle x, z \rangle}}$, where $z$ is sampled from an unnormalized density $w^a(z) = (1 + a \cdot (2\pi)^{2m} \|z\|^{2m})^{-1}$ on $\mathbb{R}^d$. This immediately suggest that if we can sample from $w^a(z)$, then we can approximate $k^a$ by summing over a finite set of samples $z_j$ instead of computing the full integral.

In fact, a similar scheme was famously previously employed in Rahimi and Recht (2007). There, it was observed that by Bochners's Theorem, Rudin (2017), any stationary kernel can be represented as $k(x, y) = \int \nu(z) e^{2\pi i \langle y, z \rangle} \cdot \overline{e^{2\pi i \langle x, z \rangle}} dz$ for some non-negative measure $\nu$. Thus, if one can sample $z_1, \ldots, z_T$ from $\nu$, one can construct an approximation

$$\hat{k}^a(x, y) = \frac{1}{T} \sum_{t=1}^{T} \cos\left(\langle z_t, x \rangle + b_t\right) \cdot \cos\left(\langle z_t, y \rangle + b_t\right), \tag{11}$$

where $b_t$ are additional i.i.d samples, sampled uniformly from $[0, 2\pi]$. In Rahimi and Recht (2007), this approximation was used as a dimension reduction for *known* analytic kernels, such as the Gaussian, for which the appropriate $\nu$ are known. Note that the samples $z_t, b_t$ can be drawn once, and subsequently used for all $x, y$ (at least in a bounded region, see the uniform approximation result in Rahimi and Recht (2007)).

For the case of interest in this paper, the SDO kernel, Bochner's representation is given by (10) in Theorem 2. Thus, to implement the sampling scheme (11) it remains to describe how one can sample from the density $w^a(z)$ on $\mathbb{R}^d$. To this end, note that $w^a(z)$ is spherically symmetric, and thus can be decomposed as $z = r\theta$, where $\theta$ is sampled uniformly from a unit sphere $S^{d-1}$ and the radius $r$ is sampled from a *one dimensional* density $u^a(r) = \frac{r^{d-1}}{1 + a(2\pi r)^{2m}}$ (see the Supplementary Material for full details on this change of variables). Next, note that sampling $\theta$ is easy. Indeed, let $g_1, \ldots, g_d$ be i.i.d standard Gaussians. Then $\theta \sim (g_1, \ldots, g_d)/\sqrt{\sum_i g_i^2}$. Thus the problem is reduced to sampling a one dimensional distribution with a single mode, with known (unnormalized) density. This can be efficiently achieved by methods such as Hamiltonian Monte Carlo (HMC). However, we found that in all cases a sufficiently fine grained discretization of the line was sufficient.

## 5 Difference between RSR and KDE Models

In this Section we construct an analytic example where the RSR estimator may differ arbitrarily from the KDE estimator with the same kernel. Thus, the models are not equivalent, and encode different prior assumptions. Briefly, we consider a block model, with two clusters. We'll show that in this particular setting, in KDE the clusters influence each other more strongly, i.e the points in one cluster contribute to the weight of the points in other cluster, yielding more uniform models. In contrast, in RSR, rather surprisingly, the density does not depend on the mutual position of the clusters (in a certain sense). Note that this is not a matter of *bandwith* of the KDE, since both models use the same kernel. We believe that this property may explain the better performance of RSR in Anomaly Detection tasks, although further investigation would be required to verify this.

Given a set of datapoints $S = \{x_i\}$, for the purposes of this section the KDE estimator is the function

$$f_{kde}(x) = f_{kde,S}(x) = \frac{1}{|S|} \sum_i k_{x_i}(x). \tag{12}$$

Let $f_{RSR}$ be the solution of (3). We will compare the ratios $f_{kde}(x_i)/f_{kde}(x_j)$ versus the corresponding quantities for RSR, $f_{RSR}^2(x_i)/f_{RSR}^2(x_j)$ for some pairs $x_i, x_j$. Note that these ratios do not depend on the normalization of $f_{kde}$ and $f_{RSR}^2$, and can be computed from the unnormalized versions. In particular, we do not require $k_{x_i}$ to be normalized in (12).

Consider a set $S$ with two components, $S = S_1 \cup S_2$, with $S_1 = \{x_1, \ldots, x_N\}$ and $S_2 = \{x'_1, \ldots x'_M\}$ and with the following kernel values:

$$K = \begin{cases} k(x_i, x_i) = k(x'_j, x'_j) = 1 & \text{for all } i \leq N, j \leq M \\ k(x_i, x_j) = \gamma^2 & \text{for } i \neq j \\ k(x'_i, x'_j) = \gamma'^2 & \text{for } i \neq j \\ k(x_i, x'_j) = \beta\gamma\gamma' & \text{for all } i, j \end{cases} \tag{13}$$

This configuration of points is a block model with two components, or two clusters. The correlations between elements in the first cluster are $\gamma^2$, and are $\gamma'^2$ in the second cluster. Inter-cluster correlations are $\beta\gamma\gamma'$. We assume that $\gamma, \gamma, \beta \in [0, 1]$ and w.l.o.g take $\gamma > \gamma'$. While this is an idealized scenario to allow analytic computations, settings closely approximating the configuration (13) often appear in real data. See Section 5.1 for an illustration. In particular, Figure 1 show a two cluster configuration in that data, and the distribution of $k(x, x')$ values.

The KDE estimator for $K$ is simply

$$f_{kde}(x_t) = \frac{1}{N+M} \left[ 1 + (N-1)\gamma^2 + M\beta\gamma\gamma' \right] \approx \frac{N}{N+M}\gamma^2 + \frac{M}{N+M}\beta\gamma\gamma', \tag{14}$$

for $x_t \in S_1$, where the second, approximate equality, holds for large $M, N$. To simplify the presentation, we shall use this approximation. However, all computations and conclusions also hold with the precise equality. For $x'_t \in S_2$ we similarly have $f_{kde}(x'_t) \approx \frac{N}{N+M}\beta\gamma\gamma' + \frac{M}{N+M}\gamma'^2$, and when $M = N$, the density ratio is

$$\frac{f_{kde}(x_t)}{f_{kde}(x'_t)} = \frac{\gamma^2 + \beta\gamma\gamma'}{\gamma'^2 + \beta\gamma\gamma'}. \tag{15}$$

The derivation of the RSR estimator is considerably more involved. Here we sketch the argument, while full details are given in Supplementary Material Section F. First, recall from the previous section that the natural gradient in the $\alpha$ coordinates is given by $2\left(\beta - N^{-1}(K\beta)^{-1}\right)$. Since the optimizer of (3) must satisfy $\nabla_f L = 0$, we are looking for $\beta \in \mathbb{R}^{N+M}$ such that $\beta = (K\beta)^{-1}$ (the term $N^{-1}$ can be accounted for by renormalization). Due to the symmetry of $K$ and since the minimizer is unique, we may take $\beta = (a, \ldots, a, b, \ldots, b)$, where $a$ is in first $N$ coordinates and $b$ is in the next $M$. Then $\beta = (K\beta)^{-1}$ is equivalent to $a, b$ solving the following system:

$$\begin{cases} a & = a^{-1}\left[1 + (N-1)\gamma^2\right] + b^{-1}M\beta\gamma\gamma' \\ b & = a^{-1}N\beta\gamma\gamma' + b^{-1}\left[1 + (M-1)\gamma'^2\right] \end{cases} \tag{16}$$

This is a non linear system in $a, b$. However, it turns out that it may be explicitly solved, up to a knowledge of a certain sign variable (see Proposition 9). Moreover, for $M = N$, the dependence on that sign variable vanishes, and we obtain

**Proposition 3.** *Consider the kernel and point configuration described by* (13)*, with $M = N$. Then for every $x_t \in S_1, x'_s \in S_2$,*

$$\frac{f_{RSR}(x_t)}{f_{RSR}(x'_s)} = \frac{\gamma^2}{\gamma'^2}. \tag{17}$$

*In particular, the ratio does not depend on $\beta$.*

It remains to compare the ratio (17) to KDE's ratio (15). If $\beta = 0$, when the clusters are maximally separated, the ratios coincide. However, let us consider the case, say, $\beta = \frac{1}{2}$, and assume that $\gamma' \ll \gamma$. Then in the denominator of (15) the larger term is $\beta\gamma\gamma'$, which comes from the influence of the first cluster on the second. This makes the whole ratio to be of the order of a constant. On the other hand, in RSR there is no such influence, and the ratio (17) may be arbitrarily large. We thus expect the gap between the cluster densities to be larger for RSR, which is indeed the case empirically. One occurence of this on real data is illustrated in Figure 1.

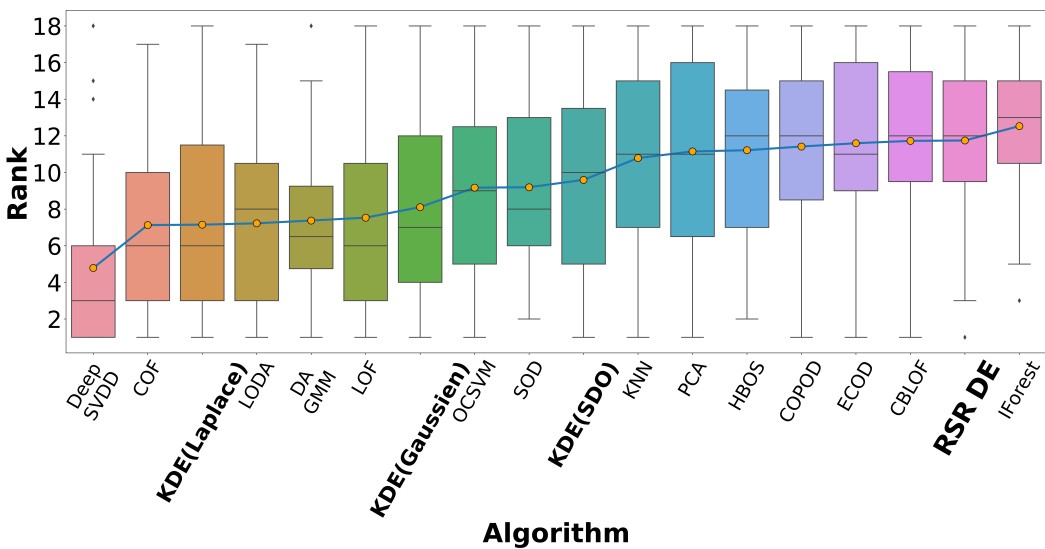

Figure 2: Anomaly Detection Results on ADBench. Relative Ranking Per Dataset, Higher is Better. RSR is Second Best Among 18 Algorithms

## 5.1 EVALUATION OF THE DIFFERENCE BETWEEN RSR AND KDE FOR REAL DATA

We have performed spectral clustering of the "letter" dataset from ADBech (Han et al. (2022)), using the empirical SDO kernel as affinity matrix for both RSR and KDE. We then have chosen two clusters that most resemble the two block model (13) in Section 5. The kernel values inside and between the clusters are shown in Figure 1a. Next, we train the RSR and KDE models for just these two clusters (to be compatible with the setting of Section 5. The results are similar for densities trained on full data). The log of these RSR and KDE densities in shown in Figure 1b (smoothed by running average). By adding an appropriate constant, we have arranged that the mean of both log densities is 0 on the first cluster. Then one can clearly see that the gap between the values on the first and second cluster is larger for the RSR model, yielding a less uniform model, as expected from the theory.

## 6 EXPERIMENTS

In this section we present the evaluation of RSR on the ADBench anomaly detection benchmark, empirically test the advantage of natural gradient descent for maintaining a non-negative $f$, and compare the likelihoods of RSR and SDO based KDE, illustrating the result of Section 5.

## 6.1 ANOMALY DETECTION RESULTS FOR ADBENCH

This section presents an evaluation of our approach on real-world tasks and data, focusing on Anomaly Detection (AD) where normalized density is not a concern. AD was chosen for evaluation due to its inherent attributes that align closely with density estimation, including the differentiation of samples from the latent and out-of-distribution. We compare our results to a golden standard AD benchmark, ADbench (Han et al. (2022)), that evaluates a wide range of 15 AD algorithms on over 47 labeled datasets. In addition, we evaluate KDE using both Gaussian and Laplace kernels, and as an ablation study, we compare RSR to KDE with SDO kernel.

We focus on the unsupervised setup, in which no labeled anomalies are given in the training phase. For all density-based approaches, we employ the negative of the density as the 'anomaly score'. The ADbench paper evaluates success on each dataset using AUC-ROC. In addition to AUC-ROC, we also focus on a ranking system as follows: for each dataset, we convert raw AUC-ROC scores of the methods into rankings from 1 to 18. Here, 18 denotes the best performance on a given dataset, and 1 the worst. This mitigates bias inherent in averaging AUC-ROC scores themselves across datasets, due to generally higher AUC-ROC scores on easier datasets. This is important since no single AD method consistently outperforms others in all situations, as discussed in detail in Han et al. (2022).

For both AUC-ROC and rank evaluations, **RSR emerges as the 2nd best AD method overall**. Notably, this achievement is with the 'vanilla' version of our method, without any pre or post-processing dedicated to AD. In contrast, many other methods are specifically tailored for AD and include extensive pre and post-processing. In Figure 2, for each algorithm we present the box plot with the average ranking over all datasets (along with quantiles). The algorithms are sorted by the

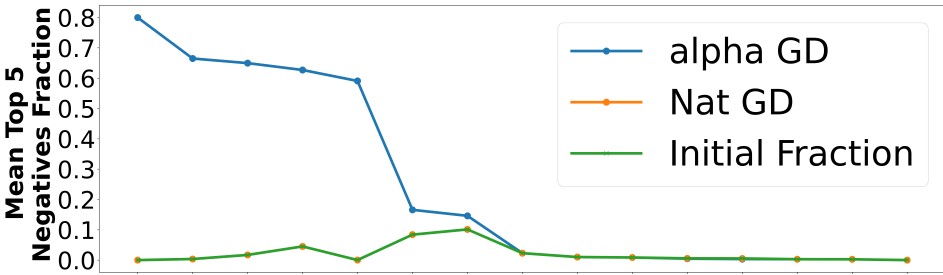

Figure 3: Fraction of negative values for natural versus $\alpha$ gradient-based optimization across datasets. The X-axis represents datasets from ADbench (see Supplementary Material Section K for details).

average ranking. A similar plot for raw AUC-ROC values is given in the supplementary material, and it presents a similar picture. As for computational cost, the entire set of 47 datasets was processed in 186 minutes using a single 3090RTX GPU and one CPU, averaging about 4 minutes per dataset.

In addition to performing well on the standard ADBench benchmark, and perhaps even more impressively, RSR excels also on the more demanding setup of *duplicate anomalies*, which was also extensively discussed in (Han et al., 2022). Here, **RSR rises to the forefront as the top AD method** (with an average AUC-ROC of 71.6 for X5 duplicates - a lead of 4% over the closest contender). This scenario, which is analogous to numerous practical situations such as equipment failures, is a focal point for ADbench's assessment of unsupervised anomaly detection methods due to its inherent difficulty, leading to substantial drops in performance for former leaders like Isolation Forest. More detailed explanations are available in the Supplementary Material.

## 6.2 FISHER-DIVERGENCE BASED HYPERPARAMETER TUNING

As noted in Section 1, dealing with unnormalized densities adds a layer of complexity to hyperparameter tuning since it prevents the use of maximum likelihood for determining optimal parameters. Consequently, for tuning the smoothness parameter $a > 0$ in both RSR and KDE with SDO kernel we employ a score-based approach. This approach measures the Fisher Divergence (FD) between a dataset sampled from an unknown distribution and a proposed distribution. Specifically, in our case, the FD between the density learned on the training set and the density inferred on the test set. Hence, the hyperparameters tuning procedure simply picks the parameters that resolve with the lowest FD. A thorough explanation of this approach, addressing important computational aspects and featuring a dedicated algorithm figure, can be found in Supplementary Material Section H.

## 6.3 NATURAL-GRADIENT VS STANDARD GRADIENT COMPARISON

We conduct an experiment to demonstrate that the standard gradient descent may significantly amplify the fraction of negative values in a solution, while the natural gradient keeps it constant. See also the discussion in Section 3.1. We have randomly chosen 15 datasets from ADBench, and for each dataset we have used 50 non negative $\alpha$ initializations. Then we have run both algorithms for 1000 iteartions. The fraction of negative values of $f_\alpha$ (on the train set) was measured at initialization, and in the end of each run. In Figure 3, for each dataset and for each method, we show an average of the highest 5 fractions among the 50 initializations. Thus, for instance, for the 'shuttle' data, the initial fraction is negligible, and is unchanged by the natural gradient. However, the standard gradient ("alpha GD" in the Figure, blue) yields about 70% negative values in the 5 worst cases (i.e. 10% of initializations).

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

## A    Overview Of The Supplementary Material

This Supplementary Material is organized as follows:

- Some basic properties of the RSR objective and of the related minimization problem: Section B

- Derivation of the Gradients of the RSR objective: Section C

- Invariance of the non-negative function cone under natural gradient steps: Section G

- Proofs related to the RSR vs KDE comparison: Section F

- Derivation of the integral form of SDO kernel, proof of Theorem 3: Section D.1

- Additional details on sampling approximation procedure: Section D.2

- Details on the Fisher Divergence estimation for RSR Hyperparameter tuning : H

- Comparison of the raw AUC-ROC metric on ADBench data: Section I.1

- Discussion of an additional test regime, with duplicated anomalies: Section I.2

- On-the-fly hyperparameter tuning procedure that was used to save time by finding the first *stable* local minimum: Section I.3

The code used for all the experiments in the paper will be publicly released with the final version of the paper.

## B    Basic Minimizer Properties

As discussed in Section 3.1, the minimizer of the RSR objective (3) always has $\mathcal{H}$ norm 1. In addition, there is no added value in multiplying the norm by a regularization scalar, since this only rescales the solution. Below we prove these statements.

**Lemma 4.** *Define*

$$f = \underset{h \in \mathcal{H}}{\operatorname{argmin}} -\frac{1}{N} \sum_i \log h^2(x_i) + \|h\|_{\mathcal{H}}^2. \tag{18}$$

*Then $f$ satisfies $\|f\|^2 = 1$. Moreover, if*

$$f' = \underset{h \in \mathcal{H}}{\operatorname{argmin}} -\frac{1}{N} \sum_i \log h^2(x_i) + \lambda^2 \|h\|_{\mathcal{H}}^2, \tag{19}$$

*for some $\lambda > 0$, then $f' = \lambda^{-1} f$.*

*Proof.* For any $h \in \mathcal{H}$ and $a > 0$,

$$\underset{a>0}{\operatorname{argmin}} -\frac{1}{N} \sum_i \log(ah)^2(x_i) + \|ah\|^2 = \tag{20}$$

$$\underset{a}{\operatorname{argmin}} -\frac{1}{N} \sum_i \log h^2(x_i) - \log a^2 + a^2 \|h\|^2. \tag{21}$$

Taking derivative w.r.t $a$ we have

$$-\frac{2a}{a^2} + 2a \|h\|^2 = 0. \tag{22}$$

Thus optimal $a$ for the problem (3) must satisfy $\|ah\|^2 = a^2 \|h\|^2 = 1$. To conclude the proof of the first claim, choose $h$ in (20) to be the minimizer in (18), $h = f$. Note that if $\|f\|_{\mathcal{H}} \neq 1$, then we can choose $a = \|f\|_{\mathcal{H}}^{-1} \neq 1$ to further decrease the value of the objective contradicting the fact that $f$ is the minimizer.

For the second claim, denoting $g = \lambda h$,

$$\underset{h \in \mathcal{H}}{\operatorname{argmin}} -\frac{1}{N} \sum_i \log h^2(x_i) + \lambda^2 \|h\|^2 \tag{23}$$

$$= \lambda^{-1} \underset{g \in \lambda \mathcal{H} = \mathcal{H}}{\operatorname{argmin}} -\frac{1}{N} \sum_i \log g^2(x_i) + \|g\|^2 + \frac{1}{N} \sum_i \log \lambda^2 \tag{24}$$

$$= \lambda^{-1} \underset{g \in \mathcal{H}}{\operatorname{argmin}} -\frac{1}{N} \sum_i \log g^2(x_i) + \|g\|^2 \tag{25}$$

$$= \lambda^{-1} f. \tag{26}$$

$\square$

## C    DERIVATION OF THE GRADIENTS, PROOF OF LEMMA 1

In this section we derive the expressions for standard and the natural gradients of the objective (5), as given in Lemma 1.

*Proof Of Lemma 1.* We first derive the expression for $\nabla_\alpha L$ in (6). Recall that $\|f\|_{\mathcal{H}}^2 = \langle \alpha, K\alpha \rangle_{\mathbb{R}^N}$ for $\alpha \in \mathbb{R}^N$, where $K_{ij} = k(x_i, x_j)$. This follows directly from the form (4), and the fact that $\langle k_x, k_y \rangle = k(x, y)$ for all $x, y \in \mathcal{H}$, by the reproducing property. For this term we have $\nabla_\alpha \langle \alpha, K\alpha \rangle = 2K\alpha$. Next, similarly by using (4), $\nabla_\alpha f(x) = (k(x_1, x), \ldots, k(x_N, x))$ for every $x \in \mathbb{R}^d$. Finally, we have

$$\nabla_\alpha \frac{1}{N} \sum_{i=1}^N \log f^2(x_i) = \frac{1}{N} \sum_{i=1}^N f^{-2}(x_i) \cdot 2f(x_i) \cdot \nabla_\alpha f(x_i) \tag{27}$$

$$= 2\frac{1}{N} \sum_{i=1}^N f^{-1}(x_i) \cdot \nabla_\alpha f(x_i) \tag{28}$$

$$= 2\frac{1}{N} K(f(x_1), \ldots, f(x_N))^{-1} \tag{29}$$

$$= 2\frac{1}{N} K (K\alpha)^{-1}. \tag{30}$$

This yields (6).

For $\nabla_f L$, we similarly have $\nabla_f \|f\|_{\mathcal{H}}^2 = 2f$, as discussed in section 3.2. Moreover,

$$\nabla_f \frac{1}{N} \sum_{i=1}^N \log f^2(x_i) = \nabla_f \frac{1}{N} \sum_{i=1}^N \log \langle f, x_i \rangle_{\mathcal{H}}^2 \tag{31}$$

$$= \frac{1}{N} \sum_{i=1}^N \langle f, x_i \rangle_{\mathcal{H}}^{-2} \cdot 2 \langle f, x_i \rangle_{\mathcal{H}} \cdot \nabla_f \langle f, x_i \rangle_{\mathcal{H}} \tag{32}$$

$$= 2\frac{1}{N} \sum_{i=1}^N \langle f, x_i \rangle_{\mathcal{H}}^{-1} x_i. \tag{33}$$

This completes the proof. $\square$

## D    SDO KERNEL DETAILS

In Section D.1 we provide a full proof of Theorem 2, while Section D.2 contains additional details on the sampling approximation of SDO.

### D.1 SDO KERNEL DERIVATION

We will prove a claim that is slightly more general than Theorem 2. For a tuple $\bar{a} \in \mathbb{R}_+^m$, define the norm

$$\|f\|_{\bar{a}}^2 = \sum_{l=0}^m a_l \sum_{|\kappa|_1=l} \frac{l!}{\kappa!} \|(D^\kappa f)\|_{L_2}^2, \tag{34}$$

where $D^\kappa$ are the $\kappa$-indexed derivative, as discussed in Section 4.1. The SDO norm is a special case with $a_0 = 1$, $a_m = a$, and $a_l = 0$ for $0 < l < m$. Let $\mathcal{H}^{\bar{a}}$ be the subspace of $L_2$ of functions with finite norm,

$$\mathcal{H}^{\bar{a}} = \{f \in L_2 \mid \|f\|_{\bar{a}} < \infty\} \tag{35}$$

and let the associated inner product be denoted by

$$\langle f, g \rangle_{\bar{a}} = \sum_{l=0}^m a_l \sum_{|\kappa|_1=l} \frac{l!}{\kappa!} \langle (D^\kappa f), (D^\kappa g) \rangle_{L_2}. \tag{36}$$

Define the Fourier transform

$$\mathcal{F}f(z) = \int_{\mathbb{R}^d} f(u)e^{-2\pi i \langle z, u \rangle} du, \tag{37}$$

and recall that we have (see for instance Stein and Weiss (1971), Grafakos (2008))

$$\mathcal{F}(D^\kappa f)(z) = \left( \prod_{j=1}^d (2\pi i z_j)^{\kappa_j} \right) \mathcal{F}f(z) \text{ for all } z \in \mathbb{R}^d. \tag{38}$$

The following connection between the $L_2$ and the derivative derived norms is well known for the standard Sobolev spaces ((Williams and Rasmussen, 2006; Saitoh and Sawano, 2016; Novak et al., 2018)). However, since (34) somewhat differs from the standard definitions, we provide the argument for completeness.

**Lemma 5.** *Set for $z \in \mathbb{R}^d$*

$$v_{\bar{a}}(z) = \left( 1 + \sum_{l=1}^m a_l \cdot (2\pi)^{2l} \|z\|^{2l} \right)^{\frac{1}{2}}. \tag{39}$$

*Then for every $f \in \mathcal{H}^{\bar{a}}$ we have*

$$\|f\|_{\bar{a}}^2 = \|v_{\bar{a}}(z) \cdot \mathcal{F}[f]\|_{L_2}^2. \tag{40}$$

*Proof.*

$$\|f\|_{\bar{a}}^2 = \sum_{l=0}^{m} a_l \sum_{|\kappa|_1=l} \frac{l!}{\kappa!} \|D^\kappa f\|_{L_2}^2 \tag{41}$$

$$= \sum_{l=0}^{m} a_l \sum_{|\kappa|_1=l} \frac{l!}{\kappa!} \|\mathcal{F}[D^\kappa f]\|_{L_2}^2 \tag{42}$$

$$= \int dz \left[ \sum_{l=0}^{m} a_l \sum_{|\kappa|_1=l} \frac{l!}{\kappa!} |\mathcal{F}[D^\kappa f](z)|^2 \right] \tag{43}$$

$$= \int dz \left[ |\mathcal{F}[f](z)|^2 + \sum_{l=1}^{m} a_l \sum_{|\kappa|_1=l} \frac{l!}{\kappa!} \left( \prod_{j=1}^{d} (2\pi z_j)^{2\kappa_j} \right) |\mathcal{F}[f](z)|^2 \right] \tag{44}$$

$$= \int dz \, |\mathcal{F}[f](z)|^2 \left[ 1 + \sum_{l=1}^{m} a_l \sum_{|\kappa|_1=l} \frac{l!}{\kappa!} \left( \prod_{j=1}^{d} (2\pi z_j)^{2\kappa_j} \right) \right] \tag{45}$$

$$= \int dz \, |\mathcal{F}[f](z)|^2 \left[ 1 + \sum_{l=1}^{m} a_l \cdot (2\pi)^{2l} \sum_{|\kappa|_1=l} \frac{l!}{\kappa!} \prod_{j=1}^{d} z_j^{2\kappa_j} \right] \tag{46}$$

$$= \int dz \, |\mathcal{F}[f](z)|^2 \left[ 1 + \sum_{l=1}^{m} a_l \cdot (2\pi)^{2l} \|z\|^{2l} \right] \tag{47}$$

$$\square$$

Using the above Lemma, the derivation of the kerenl is standard. Suppose $k^{\bar{a}}$ is the kernel corresponding to $\|f\|_{\bar{a}}$ on $\mathcal{H}^{\bar{a}}$. It remains to observe that by the reproducing property and by Lemma 5, for all $x \in \mathbb{R}^d$

$$f(x) = \langle f, k_x^{\bar{a}} \rangle_{\bar{a}} \tag{48}$$

$$= \int_{\mathbb{R}^d} dz \, \mathcal{F}[f](z) \overline{\mathcal{F}[k_x^{\bar{a}}](z)} v_{\bar{a}}^2(z). \tag{49}$$

On the other hand, by the Fourier inversion formula, we have

$$f(x) = \int dz \, \mathcal{F}[f](z) e^{2\pi i \langle x, z \rangle}. \tag{50}$$

This implies that

$$\int dz \, \mathcal{F}[f](z) e^{2\pi i \langle x, z \rangle} = \int_{\mathbb{R}^d} dz \, \mathcal{F}[f](z) \overline{\mathcal{F}[k_x^{\bar{a}}](z)} v_{\bar{a}}^2(z) \tag{51}$$

holds for all $f \in \mathcal{H}^{\bar{a}}$, which by standard continuity considerations yields

$$\mathcal{F}[k_x^{\bar{a}}](z) = \frac{e^{-2\pi i \langle x, z \rangle}}{v_{\bar{a}}^2(z)}. \tag{52}$$

Using Fourier inversion again we obtain

$$k^{\bar{a}}(x, y) = \int_{\mathbb{R}^d} \frac{e^{2\pi i \langle y-x, z \rangle}}{v_{\bar{a}}^2(z)} dz = \int_{\mathbb{R}^d} \frac{e^{2\pi i \langle y-x, z \rangle}}{1 + \sum_{l=1}^{m} a_l \cdot (2\pi)^{2l} \|z\|^{2l}} dz. \tag{53}$$

## D.2 SAMPLING APPROXIMATION

As discussed in Section 4.2, we are interested in sampling points $z \in \mathbb{R}^d$ from a finite non negative measure with density given by $w^a(z) = (1 + a \cdot (2\pi)^{2m} \|z\|^{2m})^{-1}$. With a slight overload of notation, we will also denote by $w_a$ the scalar function $w_a : \mathbb{R} \to \mathbb{R}$,

$$w^a(r) = (1 + a \cdot (2\pi)^{2m} r^{2m})^{-1}. \tag{54}$$

First, note that $w_a(z)$ depends on $z$ only through the norm $\|z\|$, and thus a spherically symmetric function. Therefore, with a spherical change of variables, we can rewrite the integrals w.r.t $w_a^{-2}$ as follows: For any $f : \mathbb{R}^d \to \mathbb{C}$,

$$\int_{\mathbb{R}^d} w_a(z) f(z) dz = \int_0^\infty dr \int_{S^{d-1}} d\theta \ w_a(r) A_{d-1}(r) f(r\theta) \tag{55}$$

$$= A_{d-1}(1) \int_0^\infty dr \int_{S^{d-1}} d\theta \ \left[ w_a(r) r^{d-1} \right] \cdot f(r\theta). \tag{56}$$

Here $S^{d-1}$ the unit sphere in $\mathbb{R}^d$, $\theta$ is sampled from the uniform probability measure on the sphere, $r$ is the radius, and

$$A_{d-1}(r) = \frac{2\pi^{d/2}}{\Gamma(d/2)} r^{d-1} \tag{57}$$

is the $d-1$ dimensional volume of the sphere or radius $r$ in $\mathbb{R}^d$. The meaning of (56) is that to sample from $w_a^{-2}$, we can sample $\theta$ uniformly from the sphere (easy), and $r$ from a density

$$\zeta(r) = w_a(r) r^{d-1} = \frac{r^{d-1}}{1 + a \cdot (2\pi)^{2m} r^{2m}} \tag{58}$$

on the real line. Note that the condition $m > d/2$ that we impose throughout is necessary. Indeed, without this condition the decay of $\zeta(r)$ would not be fast enough at infinity, and the density would not have a finite mass.

As discussed in Section 4.2, $\zeta(r)$ is a density on a real line, with a single mode and an analytic expression, which allows easy computation of the derivatives. Such distributions can be efficiently sampled using, for instance, off-the-shelf Hamiltonian Monte Carlo (HMC) samplers, Betancourt (2017). In our experiments we have used an even simpler scheme, by discretizing $\mathbb{R}$ into a grid of 10000 points, with limits wide enough to accommodate a wide range of parameters $a$.

## E    A FEW BASIC PROPERTIES OF THE KERNEL

**Proposition 6.** *The kernel* (10) *is real valued and satisfies*

$$K^a(x, y) = \int_{\mathbb{R}^d} \frac{\cos\left(2\pi \langle y - x, z \rangle\right)}{1 + a \cdot (2\pi)^{2m} \|z\|^{2m}} dz. \tag{59}$$

*Proof.* Write $e^{2\pi i \langle y-x, z \rangle} = \cos(2\pi \langle y - x, z \rangle) + i \sin(2\pi \langle y - x, z \rangle)$ and observe that $\sin$ is odd in $z$, while $1 + a \cdot (2\pi)^{2m} \|z\|^{2m}$ is even. $\qquad \square$

**Proposition 7.** *For all* $x, y \in \mathbb{R}^d$,

$$K^b(x, y) = b^{-\frac{d}{2m}} K^1(b^{-\frac{1}{2m}} x, b^{-\frac{1}{2m}} y) \tag{60}$$

*Proof.* Write $u = b^{\frac{1}{2m}} z$ and note that $du = (b^{\frac{1}{2m}})^d dz$. We have

$$K^b(x, y) = \int_{\mathbb{R}^d} \frac{\cos\left(2\pi \langle y - x, z \rangle\right)}{1 + b \|2\pi \cdot z\|^{2m}} dz \tag{61}$$

$$= \int_{\mathbb{R}^d} \frac{\cos\left(2\pi \left\langle b^{-\frac{1}{2m}} (y - x), u \right\rangle\right)}{1 + \|2\pi u\|^{2m}} du \cdot b^{-\frac{d}{2m}} \tag{62}$$

$$= b^{-\frac{d}{2m}} K^1(b^{-\frac{1}{2m}} x, b^{-\frac{1}{2m}} y). \tag{63}$$

$$\square$$

**Lemma 8.** *There is a function $c(m)$ of $m$ such that for every $x \in \mathbb{R}^d$,*

$$\int \left( K^a(x, y) \right)^2 dy = c(m) \cdot a^{-\frac{d}{2m}}. \tag{64}$$

*Proof.* Recall that for fixed $x$ and $a$, the Fourier transform satisfies $\mathcal{F}(k_x^a)(z) = \frac{e^{-2\pi i\langle x,z\rangle}}{1+a(2\pi)^{2m}\|z\|^{2m}}$, where $k_x^a(\cdot) = K^a(x, \cdot)$ (see eq. (52)). We thus have

$$\|k_x^a\|_{L_2}^2 = \|\mathcal{F}(k_x^a)\|_{L_2}^2 \tag{65}$$

$$= \int \frac{e^{-2\pi i\langle x,z\rangle} \cdot \overline{e^{-2\pi i\langle x,z\rangle}}}{\left(1 + a(2\pi)^{2m}\|z\|^{2m}\right)^2} dz \tag{66}$$

$$= \int \frac{1}{\left(1 + a(2\pi)^{2m}\|z\|^{2m}\right)^2} dz \tag{67}$$

$$= a^{-\frac{d}{2m}} \int \frac{1}{\left(1 + (2\pi)^{2m}\|z'\|^{2m}\right)^2} dz', \tag{68}$$

$$\tag{69}$$

where we have used the variable change $z = a^{-\frac{1}{2m}}z'$. $\qquad\square$

# F   KDE VS RSR COMPARISON PROOFS

In this section we develop the ingredients required to prove Proposition 3. In section F.1 we reduce the solution of the RSR problem for the two block model to a solution of a non-linear system in two variables, and derive the solution of this system. In section F.2 we use these results to prove Proposition 3.

## F.1   SOLUTION OF RSR FOR A 2-BLOCK MODEL

As discussed in section 5, any RSR solution $f$ must be a zero point of the natural gradient, $\nabla_f L = 0$. Using the expressions given following Lemma 1, this implies $\beta = \frac{1}{N}(K\beta)^{-1}$. Since we are only interested in $f$ up to a scalar normalization, we can equivalently assume simply $\beta = (K\beta)^{-1}$. Further, by symmetry consideration we may take $\beta = (a, \ldots, a, b, \ldots, b)$, where $a$ is in first $N$ coordinates and $b$ is in the next $M$. Then, as mentioned in section 5, $\beta = (K\beta)^{-1}$ is equivalent to $a, b$ solving the following system:

$$\begin{cases} a &= a^{-1}\left[1 + (N-1)\gamma^2\right] + b^{-1}M\beta\gamma\gamma' \\ b &= a^{-1}N\beta\gamma\gamma' + b^{-1}\left[1 + (M-1)\gamma'^2\right] \end{cases} \tag{70}$$

It turns out that it is possible to derive an expression for the ratio of the squares of the solutions to this system in the general case.

**Proposition 9** (Two Variables RSR System). *Let $a, b$ be solutions of*

$$\begin{cases} a &= H_{11}a^{-1} + H_{12}b^{-1} \\ b &= H_{21}a^{-1} + H_{22}b^{-1} \end{cases} \tag{71}$$

*Then*

$$a^2/b^2 = H_{11}^2 \left( \frac{-(H_{21} + H_{12}) - \rho\sqrt{(H_{21} - H_{12})^2 + 4H_{11}H_{22}}}{2H_{11}H_{22} + H_{12}\left[-(H_{21} - H_{12}) + \rho\sqrt{(H_{21} - H_{12})^2 + 4H_{11}H_{22}}\right]} \right)^2 \tag{72}$$

*for a $\rho$ satisfying $\rho \in \{+1, -1\}$.*

*Proof.* Write $u = a^{-1}$, $v = b^{-1}$, and multiply the first and second equations by $u$ and $v$ respectively. Then we have

$$\begin{cases} 1 &= H_{11}u^2 + H_{12}uv \\ 1 &= H_{21}uv + H_{22}v^2. \end{cases} \tag{73}$$

We write

$$v = \left(1 - H_{11}u^2\right)/H_{12}u. \tag{74}$$

Also, from the first equation,

$$H_{12}uv = 1 - H_{11}u^2. \tag{75}$$

Substituting into the second equation,

$$1 = \frac{H_{21}}{H_{12}}\left(1 - H_{11}u^2\right) + H_{22}\frac{\left(1 - H_{11}u^2\right)^2}{\left(H_{12}u\right)^2}. \tag{76}$$

Finally setting $s = u^2$ and multiplying by $H_{12}^2 s$,

$$H_{12}^2 s = H_{21}H_{12}s\left(1 - H_{11}s\right) + H_{22}\left(1 - H_{11}s\right)^2. \tag{77}$$

Collecting terms, we have

$$s^2(H_{11}^2 H_{22} - H_{11}H_{12}H_{21}) + s(H_{12}H_{21} - H_{12}^2 - 2H_{11}H_{22}) + H_{22} = 0. \tag{78}$$

Solving this, we get

$$s = \frac{-(H_{12}H_{21} - H_{12}^2 - 2H_{11}H_{22}) \pm \sqrt{(H_{12}H_{21} - H_{12}^2 - 2H_{11}H_{22})^2 - 4(H_{11}^2 H_{22} - H_{11}H_{12}H_{21})H_{22}}}{2(H_{11}^2 H_{22} - H_{11}H_{12}H_{21})}. \tag{79}$$

The expression inside the square root satisfies

$$(H_{12}H_{21} - H_{12}^2 - 2H_{11}H_{22})^2 - 4(H_{11}^2 H_{22} - H_{11}H_{12}H_{21})H_{22} \tag{80}$$

$$= (H_{12}(H_{21} - H_{12}) - 2H_{11}H_{22})^2 - 4(H_{11}^2 H_{22} - H_{11}H_{12}H_{21})H_{22} \tag{81}$$

$$= H_{12}^2(H_{21} - H_{12})^2 - 4H_{11}H_{22}H_{12}(H_{21} - H_{12}) + 4H_{11}H_{12}H_{21}H_{22} \tag{82}$$

$$= H_{12}^2(H_{21} - H_{12})^2 + 4H_{11}H_{22}H_{12}^2 \tag{83}$$

$$= H_{12}^2\left[(H_{21} - H_{12})^2 + 4H_{11}H_{22}\right] \tag{84}$$

Thus, simplifying, we have

$$u^2 = s = \frac{-(H_{12}(H_{21} - H_{12}) - 2H_{11}H_{22}) + \rho H_{12}\sqrt{(H_{21} - H_{12})^2 + 4H_{11}H_{22}}}{2H_{11}(H_{11}H_{22} - H_{12}H_{21})}, \tag{85}$$

where $\rho \in \{+1, -1\}$.

Rewriting (74) again, we have

$$v^2 = \frac{\left(1 - H_{11}u^2\right)^2}{H_{12}^2 u^2}. \tag{86}$$

Further,

$$a^2/b^2 = v^2/u^2 = \frac{\left(1 - H_{11}u^2\right)^2}{H_{12}^2 u^4} \tag{87}$$

$$= \left(\frac{1 - H_{11}u^2}{H_{12}u^2}\right)^2 \tag{88}$$

$$= \left(\frac{1}{H_{12}u^2} - \frac{H_{11}}{H_{12}}\right)^2 \tag{89}$$

$$= \left(\frac{H_{11}}{H_{12}}\right)^2 \left(\frac{2(H_{11}H_{22} - H_{12}H_{21})}{-(H_{12}(H_{21} - H_{12}) - 2H_{11}H_{22}) + \rho H_{12}\sqrt{(H_{21} - H_{12})^2 + 4H_{11}H_{22}}} - 1\right)^2 \tag{90}$$

$$= \left(\frac{H_{11}}{H_{12}}\right)^2 \left(\frac{2(H_{11}H_{22} - H_{12}H_{21}) + (H_{12}(H_{21} - H_{12}) - 2H_{11}H_{22}) - \rho H_{12}\sqrt{(H_{21} - H_{12})^2 + 4H_{11}H_{22}}}{-(H_{12}(H_{21} - H_{12}) - 2H_{11}H_{22}) + \rho H_{12}\sqrt{(H_{21} - H_{12})^2 + 4H_{11}H_{22}}}\right)^2 \tag{91}$$

$$= \left(\frac{H_{11}}{H_{12}}\right)^2 \left(\frac{-2H_{12}H_{21} + H_{12}(H_{21} - H_{12}) - \rho H_{12}\sqrt{(H_{21} - H_{12})^2 + 4H_{11}H_{22}}}{-(H_{12}(H_{21} - H_{12}) - 2H_{11}H_{22}) + \rho H_{12}\sqrt{(H_{21} - H_{12})^2 + 4H_{11}H_{22}}}\right)^2 \tag{92}$$

$$= H_{11}^2 \left(\frac{-2H_{21} + H_{21} - H_{12} - \rho\sqrt{(H_{21} - H_{12})^2 + 4H_{11}H_{22}}}{-(H_{12}(H_{21} - H_{12}) - 2H_{11}H_{22}) + \rho H_{12}\sqrt{(H_{21} - H_{12})^2 + 4H_{11}H_{22}}}\right)^2 \tag{93}$$

$$= H_{11}^2 \left(\frac{-(H_{21} + H_{12}) - \rho\sqrt{(H_{21} - H_{12})^2 + 4H_{11}H_{22}}}{-(H_{12}(H_{21} - H_{12}) - 2H_{11}H_{22}) + \rho H_{12}\sqrt{(H_{21} - H_{12})^2 + 4H_{11}H_{22}}}\right)^2 \tag{94}$$

$$= H_{11}^2 \left(\frac{-(H_{21} + H_{12}) - \rho\sqrt{(H_{21} - H_{12})^2 + 4H_{11}H_{22}}}{2H_{11}H_{22} + H_{12}\left[-(H_{21} - H_{12}) + \rho\sqrt{(H_{21} - H_{12})^2 + 4H_{11}H_{22}}\right]}\right)^2 \tag{95}$$

$$\square$$

## F.2 PROOF OF PROPOSITION 3

Similarly to the case with KDE, we will use the following approximation of the system (70)

$$\begin{cases} a &= a^{-1}N\gamma^2 + b^{-1}M\beta\gamma\gamma' \\ b &= a^{-1}N\beta\gamma\gamma' + b^{-1}M\gamma'^2 \end{cases} \tag{96}$$

*Proof.* Let $f$ be the RSR solution. By definition, the ratio $\frac{f(x_t)}{f(x'_s)}$ is given by $a^2/b^2$ where $a, b$ are the solutions to (96). That is, we take $H_{12} = H_{21} = \beta\gamma\gamma'$, $H_{11} = \gamma^2$, and $H_{22} = \gamma'^2$ in Proposition 9. Note that we have removed the dependence on $N$, since it does not affect the ratio. By Proposition 9, substituting into (72),

$$H_{11}^2 \left(\frac{-(H_{21} + H_{12}) - \rho\sqrt{(H_{21} - H_{12})^2 + 4H_{11}H_{22}}}{2H_{11}H_{22} + H_{12}\left[-(H_{21} - H_{12}) + \rho\sqrt{(H_{21} - H_{12})^2 + 4H_{11}H_{22}}\right]}\right)^2 \tag{97}$$

$$= H_{11}^2 \left(\frac{-H_{12} - \rho\sqrt{H_{11}H_{22}}}{H_{11}H_{22} + H_{12}\rho\sqrt{H_{11}H_{22}}}\right)^2 \tag{98}$$

$$= \gamma^4 \left(\frac{-2\beta\gamma\gamma' - 2\rho\gamma\gamma'}{2\gamma^2\gamma'^2 + 2\beta\gamma\gamma'\rho\gamma\gamma'}\right)^2 \tag{99}$$

$$= \left(\frac{\gamma^2\gamma\gamma'}{\gamma^2\gamma'^2}\right)^2 \frac{(\beta + \rho)^2}{(1 + \beta\rho)^2} \tag{100}$$

It remains to note that $\frac{(\beta+\rho)^2}{(1+\beta\rho)^2} = 1$ for any $\beta$ and $\rho \in \{+1, -1\}$. $\qquad\square$

## G  INVARIANCE OF $\mathcal{C}$ UNDER NATURAL GRADIENT

Define the non-negative cone of functions $\mathcal{C} \subset \mathcal{H}$ by

$$\mathcal{C} = \{f \in \mathcal{H} \mid f(x) \geq 0 \ \forall x \in \mathcal{X}\}. \tag{101}$$

As discussed in section 3.1, the functional $L(f)$ is convex on $\mathcal{C}$.

We now show that if the kernel $k$ is non-negative, then the cone $\mathcal{C}$ is invariant under the natural gradient steps. In particular, this means that if one starts with initialization in $\mathcal{C}$ (easy to achieve), then the optimization trajectory stays in $\mathcal{C}$, without a need for computationally heavy projection methods. Note that this is unlikely to be true for the standard gradient. Recall that the expression (6) for the natural gradient is given in Lemma 1.

**Proposition 10.** *Assume that $k(x, x') \geq 0$ for all $x, x' \in \mathcal{X}$ and that $\lambda < 0.5$. If $f \in \mathcal{C}$, then also $f' := f - 2\lambda \left[ f - \frac{1}{N} \sum_{i=1}^{N} f^{-1}(x_i) k_{x_i} \right] \in C$.*

*Proof.* Indeed, by opening the brackets,

$$f' = (1 - 2\lambda) f + 2\lambda \left[ \frac{1}{N} \sum_{i=1}^{N} f^{-1}(x_i) k_{x_i} \right],$$

which is a non-negative combination of functions in $\mathcal{C}$, thus yielding the result. $\qquad\square$

## H  FISHER DIVERGENCE FOR HYPER-PARAMETERS SELECTION

The handling of unnormalized models introduces a particular nuance in the context of hyperparameter tuning, as it prevents the use of the maximum likelihood of the data in order to establish the optimal parameter. When confronted with difficulties associated with normalization, it is common to resort to score-based methods. The score function is defined as

$$s(x; a) = \nabla_x \log p_m(x; a), \tag{102}$$

where $p_m(x; a)$ is a possibly unnormalized probability density on $\mathbb{R}^d$, evaluated at $x \in \mathbb{R}^d$, and dependent on the hyperparameter $a$. Since the normalization constant is independent of $x$, and $s$ is defined via the gradient in $x$, $s$ is independent of the normalization. As a result, distance metrics between distributions that are based on the score function, such as the Fisher Divergence, can be evaluated using non-normalized distributions.

In this work, we employ this concept, leveraging it to identify the choice of parameters (in our case, $a$, the smoothness parameter) that minimize the FD between the density learned on the training set and the density inferred on the test set. Specifically, we apply *score-matching* (Hyvärinen and Dayan (2005)), a particular approach to measuring the Fisher divergence between a dataset sampled from an unknown distribution and a proposed distribution model. The full details of the procedure are in Supllmentry Metrial Section J

### H.1  SCORE MATCHING AND FISHER DIVERGENCE

Given independent and identically distributed samples $x_1, \ldots, x_N \in \mathbb{R}^D$ from a distribution $p_d(x)$ and an un-normalized density learned, $\tilde{p_m}(x; a)$ (where $a$ is a parameter). Score matching sets out to reduce the Fisher divergence between $p_d$ and $\tilde{p_m}(\cdot; a)$, formally expressed as

$$L(a) = \frac{1}{2} \cdot E_{p_d}[\|s_m(x; a) - s_d(x)\|^2]$$

As detailed in Hyvärinen and Dayan (2005), the technique of integration by parts can derive an expression that does not depend on the unknown latent score function $s_d$:

$$L(a; x_1, \ldots, x_n) = \frac{1}{N} \sum_{i=1}^{N} \left[ tr(\nabla_x s_m(x_i; a)) + \frac{1}{2} \cdot \|s_m(x_i; a)\|^2 \right] + C$$

In this context, C is a constant independent of $a$, $tr(\cdot)$ denotes the trace of a matrix, and $\nabla_x s_m(x_i; a) = \nabla_x^2 log(\tilde{p_m}(x_i; a))$ is the Hessian of the learned log-density function evaluated at $x_i$.

## H.2 THE HESSIAN ESTIMATION FOR SMALL $a$'S

Deriving the Hessian for small $a$ values proves to be challenging. Note that small $a$ values signify overfitting to the training data, consequently, this leads to a density that is mainly close to zero between samples, thereby making the process highly susceptible to significant errors in numerically calculating the derivatives. This situation results in a Hessian that is fraught with noise. Hence, our strategy focuses on locating a stable local minimum with the highest possible $a$. In this context, we define a stable local minimum as a point preceded and succeeded by three points, each greater than the focal point.

## H.3 APPROXIMATING THE HESSIAN TRACE

Although this method holds promise, it's worth noting the computational burden tied to the calculation of the Hessian trace. To mitigate this, we rely on two techniques. First, we utilize Hutchinson's trace estimator (Hutchinson, 1989), a robust estimator that facilitates the estimation of any matrix's trace through a double product with a random vector $\epsilon$:

$$Tr(H) = E_\epsilon \left[ \epsilon^T H \epsilon \right].$$

Here $\epsilon$ is any random vector on $\mathbb{R}^d$ with mean zero and covariance $I$. This expression allows to reduce amount of computation of $Tr(H)$, by computing the products $H\epsilon$ directly, for a few samples of $\epsilon$, without the need to compute the full $H$ itself. A similar strategy has been recently employed in Grathwohl et al. (2018) in a different context, for a trace computation of a Jacobian of a density transformation, instead of the score itself.

In more detail, score computations can be performed efficiently and in a 'lazy' manner using automatic differentiation, offered in frameworks such as PyTorch. This allows us to compute a vector-Hessian product $H\epsilon$ per sample without having to calculate the entire Hessian for all samples, a tensor of dimensions $N \times (d \times d)$, in advance. More specifically, we utilize PyTorch's automatic differentiation for computing the score function, which is a matrix of $N \times d$. Subsequently, this is multiplied by $\epsilon$. We then proceed with a straightforward differentiation $\nabla_x s(x_i) = \frac{1}{h} \cdot (s(x_i + h \cdot \epsilon) - s(x_i))$ for small step $h$, followed by a summation which is lazily calculated through PyTorch (see Algorithm 1).

---

**Algorithm 1:** Calculating Hutchinson's Trace Estimator

**Require:** Score function $s$, small constant $h$, sample $x$, # of random vectors $n$
 1: Initialize $traceEstimator$ to 0
 2: **for** $i = 1$ to $n$ **do**
 3:     Sample random vector $\epsilon$ from normal distribution
 4:     Calculate $s(a; x + h * \epsilon)$
 5:     Calculate $(s(a; x + h * \epsilon) - s(a; x))$
 6:     Compute $(1/h) \cdot (s(a; x + h * \epsilon) - s(a; x)) \cdot \epsilon$
 7:     Add result to $traceEstimator$
 8: **end for**
 9: Return $\frac{traceEstimator}{n}$

---

## I  EXPERIMENTS

### I.1  AUC-ROC PERFORMANCE ANALYSIS

In Section 6.1 we presented a comparative study of various methods according to their ranking across different datasets. This section provides an analysis of the raw AUC-ROC values themselves.

We present two figures to elucidate our findings. Figure 4a is similar to the box plot from shown in Figure 2 in the main text but includes raw AUCROC values instead of rankings. As the figure

shows, RSR also secures the second-highest position in terms of average raw AUC-ROC (second only to IForest), as represented by the order of methods on the X-axis. Furthermore, the median line within the boxes indicates that RSR is the method with the highest median score. Figure 4b shows a heatmap representation of the AUC-ROC values. In this visualization, the size of the circle symbolizes the corresponding AUC value, while the color gradient signifies the deviation in AUC value from RSR. The purpose of this heatmap is to offer a graphical interpretation of the AUC-ROC performance levels, demonstrating how they diverge from the performance of RSR.

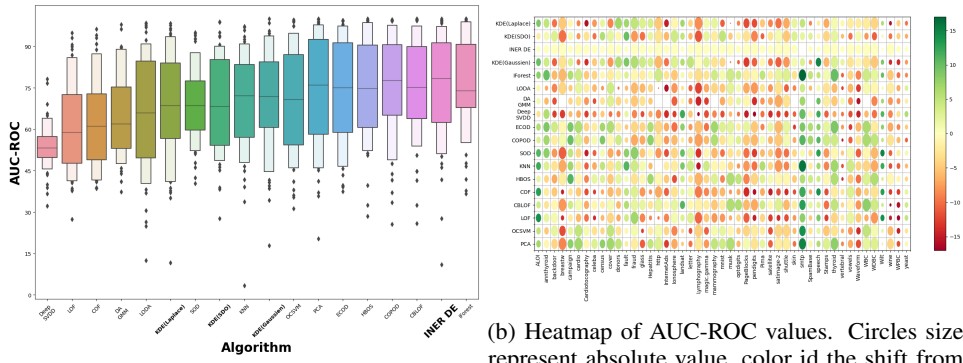

(a) Box plot presenting actual AUC-ROC values.

(b) Heatmap of AUC-ROC values. Circles size represent absolute value, color id the shift from RSR.

## I.2 DUPLICATE ANOMALIES.

Duplicate anomalies are often encountered in in various applications due to factors such as recording errors Kwon et al. (2018), a circumstance termed as "anomaly masking" (Campos et al., 2016; Guha et al., 2016), posing significant hurdles for diverse AD algorithms. The significance of this factor is underscored in ADbench (Han et al., 2022), where duplicate anomalies are regarded as the most difficult setup for anomaly detection, thereby attracting considerable attention. To replicate this scenario, ADbench duplicates anomalies up to six times within training and test sets, subsequently examining the ensuing shifts in AD algorithms' performance across the 47 datasets discussed in our work.

As shown in ADBench, unsupervised methods are considerably susceptible to repetitive anomalies. In particular, as shown in Fig. 7a in the main text there, performance degradation is directly proportional to the increase in anomaly duplication. With anomalies duplicated six times, unsupervised methods record a median AUC-ROC reduction of -16.43%, where in RSR the drop is less then 2%. This universal performance deterioration can be attributed to two factors: 1) The inherent assumption made in these methods that anomalies are a minority in the dataset, a presumption crucial for detection. The violation of this belief due to the escalation in duplicated anomalies triggers the noticed performance downturn. 2) Methods based on nearest neighbours assume that anomalies significantly deviate from the norm (known as "Point anomalies"). However, anomaly duplication mitigates this deviation, rendering the anomaly less distinguishable. Notice that while the first factor is less of a problem for a density based AD algorithm ( any time the anomalies are still not the major part of the data), the second factor could harmful to DE based AD algorithms as well. The evidence of RSR possessing the highest median and its robustness to duplicate anomalies, along with a probable resistance to a high number of anomalies, not only emphasizes its superiority as a DE method but also underscores its potential to serve as a state-of-the-art AD method.

## I.3 HYPERPARAMETER TUNING WITH ON-THE-FLY FISHER-DIVERGENCE MINIMIZATION

A primary task at hand involves hyperparameter tuning to select the optimal $a$ according to the Fisher-Divergence (FD) procedure, as detailed in section H and Section 6.1. For efficient computation, we employ an on-the-fly calculation approach. The process initiates with a $a$ value that corresponds to the 'order'-th largest tested point. Subsequently, we explore one larger and one smaller $a$ value. If both these points exceed the currently tested point, we continue sampling. Otherwise, we shift the tested point to a smaller value. To avoid redundant calculations, the results are continually stored, ensuring

each result is computed only once. This methodology provides a balance between computational efficiency and thorough exploration of the hyperparameter space.

## J  Algorithm Overview

In this section, we outline the procedure for our density estimation algorithm. Let $X \in \mathbb{R}^{N \times d}$ represent the training set and $Y \in \mathbb{R}^{N \times d}$ denote the test set for which we aim to compute the density. Initially, we establish the following hyper-parameters: $1.N_z$, the number of samples $Z$ taken for the kernel estimation. (Notice $N_z$ is $T$ from (11)) $2.n_{iters}$, the number of iterations used in the gradient decent approximating the optimization for $\alpha$. $3.lr$, the learning rate applied in the gradient decent $\alpha$ optimization process. $4.n_{fd\_iters}$, the number of iterations for the Hessian trace approximation. $5.h$, the step size employed for the Hessian trace. Then, we follow Algorithm 2.

---

**Algorithm 2:** Density Estimation Procedure with Hypeparameter Tuning

---

1: **repeat**
2:   Estimate the SDO kernel matrix via sampling as described in Section 3 and detailed in Algorithm 3.
3:   Determine the optimal $\alpha$ as described in Section 3 and detailed in Algorithm 4. $optimal\_\alpha$ forms $f := f^* \triangleq f_{optimal\_\alpha}$ and the density estimator is $F^2 \triangleq (f)^2$.
4:   Compute $F^2(Y)$ the density over $Y$ as described in Section 3 and detailed in Algorithm 5.
5:   Assess the Fisher-divergence as described in Supplementary Material Section H and detailed in both Algorithm 6 and Algorithm 1.
6: **until** For each smoothness parameter $a$.
7: The density estimator $F^2$ corresponds to the $a$ value that yields the minimal FD.

---

---

**Algorithm 3:** $K(\cdot) \rightarrow \mathbb{R}^{2 \times N}$ : Sampling the Multidimensional SDO Kernel

---

**Require:** $X, Y, a$
1: $\boldsymbol{\theta} \leftarrow \frac{\boldsymbol{g}}{\|\boldsymbol{g}\|_2} ; \boldsymbol{g} \sim \mathcal{N}(\mathbf{0}, \mathbf{1})$
2: $\boldsymbol{r} \sim \frac{r^{d-1}}{1+a(2\pi r)^{2m}}$ using grid search.
3: $Z \leftarrow \boldsymbol{r} \cdot \boldsymbol{\theta}^T$
4: $\boldsymbol{b} \sim U[0, 2\pi]$
5: **return** $\frac{1}{N_z} \cdot \cos(X \cdot Z + b) \cdot (\cos(Y \cdot Z + b))^T$

---

---

**Algorithm 4:** $Optimal\_alpha(\cdot) \rightarrow \mathbb{R}^N$ : Calculating the Optimal Alphas

---

**Require:** $X, \text{lr}, n_{\text{iters}}$
1: $K \leftarrow K(X, X)$
2: $\alpha \leftarrow [|\alpha_0|, \dots, |\alpha_N|] : \alpha_i \sim \mathcal{N}(0, 1)$
3: **for** $i = 1$ **to** $n_{\text{iters}}$ **do**
4:   $\alpha \leftarrow \alpha - 2 \cdot lr \cdot ((K \cdot \alpha) - (K \cdot (1./(K \cdot \alpha)))/N_{\text{data}})$
5: **end for**
6: **return** $\alpha$

---

## K  Figure 3 - List of Datasets

Those are the datasets for which we compared the fractions of negative values for $alpha$ optimization vs natural gradients presented in Figure 3, ordered according to the X-axis in the figure :

1. Mnist
2. Shuttle
3. PageBlocks

---

**Algorithm 5:** $F^2(\cdot) \to \mathbb{R}^{N_Y}$ : Density over coordinates $Y$ given observations $X$.

---

**Require:** $X, Y$
1: $\alpha \leftarrow Optimal\_alpha(X, \ldots)$
2: $e \leftarrow K(X, Y)$
3: **return** $(e \cdot \alpha)^2$

---

---

**Algorithm 6:** $FD(\cdot) \to \mathbb{R}$ : Fisher Divergence with Hessian Trace Approximation.

---

**Require:** $X$(Train set), $Y$(Test set), $n_{fd\_iters}, h$
1: $scores \leftarrow \nabla \sum \log(F^2(X, Y))$
2: traces_sum $\leftarrow \mathbf{0}$
3: **for** $i = 1$ **to** $n_{\text{iters}}$ **do**
4:    $\epsilon \sim \mathcal{U}(\{-1, 0, 1\}^{\text{size}(X)})$
5:    shifted_scores $\leftarrow \nabla \sum \log(F^2(X, Y + h \cdot \epsilon))$
6:    traces_sum $\leftarrow$ traces_sum $+ \sum((\text{shifted\_scores} - scores) \cdot \epsilon)/h$
7: **end for**
8: traces $\leftarrow$ traces_sum$/n_{\text{iters}}$
9: **return** $\mathbb{E}[\text{traces} + \frac{1}{2} \cdot \|\mathbf{scores}\|^2]$

---

4. Mammography

5. Magic.gamma

6. Skin

7. Backdoor

8. Glass

9. Lymphography

10. Stamps

11. WDBC

12. SpamBase

13. Hepatites

14. Wine

15. Letter

## L   CONSISTENCY THEOREM

In this Section we state and prove the consistency result for the RSR estimators.

Recall that for any $a > 0$, $\mathcal{H}^a$ is the RKHS with the norm given by (8) and the kernel $k^a(x, y)$ given by (10).

For any $f \in \mathcal{H}^1$ define the RSR loss

$$L(f) = L^a(f) = \frac{1}{2}\left(-\frac{1}{N}\sum \log f^2(x_i) + \|f\|_{\mathcal{H}^a}^2\right). \tag{103}$$

Note that $f \in \mathcal{H}^1$ if and only if $f \in \mathcal{H}^a$ for every $a > 0$. Recall from the discussion in Section 3.2 and that $L$ is convex when restricted to the open cone

$$\mathcal{C}' = (\mathcal{C}^a)' = \left\{f \in span\,\{k_{x_i}\}_1^N \;\mid\; f(x_i) > 0\right\}. \tag{104}$$

Note that $\mathcal{C}'$ depends on $a$ since the kernel $k_{x_i} = k_{x_i}^a$ depends on $a$. Observe also that compared to Section 3.2, here we require only positivity on the data points $x_i$ rather than on all $x \in \mathbb{R}^d$, and we restrict the cone to the span of $x_i$ since all the solutions will be obtained there in any case. This of course does not affect the convexity.

The consistency result we prove is as follows:

**Theorem 11.** *Let $x_1, \ldots, x_N$ be i.i.d samples from a compactly supported density $v^2$ on $\mathbb{R}^d$, such that $v \in \mathcal{H}^1$. Set $a = a(N) = 1/N$, and let $u_N = u(x_1, \ldots, x_N; a(N))$ be the minimizer of the objective* (103) *in the cone* (104). *Then $\|u_N - v\|_{L_2}$ converges to $0$ in probability.*

In words, when $N$ grows, and the regularisation size $a(N)$ decays as $1/N$, the the RSR estimators $u_N$ converge to $v$ in $L_2$.

As discussed in the main text, note also that since $\|v\|_{L_2} = 1$ (as its a density), and since $\|u_N - v\|_{L_2} \to 0$, it follows by the triangle inequality that $\|u_N\|_{L_2} \to 1$. That is, the estimator $u_N$ becomes approximately normalized as $N$ grows.

In Section L.1 below we provide an overview of the proof, and in Section L.2 full details are provided.

### L.1   OVERVIEW OF THE PROOF

As discussed in Section 2, consistency for the 1 dimensional case was shown in Klonias (1984) and our approach here follows similar general lines. The differences between the arguments are due to the more general multi dimensional setting here, and due to some difference in assumptions.

To simplify the notation in what follows we set $\|\cdot\|_a := \|\cdot\|_{\mathcal{H}^a}$, $\langle\cdot,\cdot\rangle_a := \langle\cdot,\cdot\rangle_{\mathcal{H}^a}$. Recall that $k^a(x, y)$ denotes the kernel corresponding to $\mathcal{H}^a$,

The first step of the proof is essentially a stability result for the optimization of (103), given by Lemma 12 below. In this Lemma we observe that $L$ is strongly convex and thus for any function $v$, we have

$$\|u - v\|_{\mathcal{H}^a} \le \|\nabla L(v)\|_{\mathcal{H}^a}, \tag{105}$$

where $u$ is the true minimizer of $L$ in $\mathcal{C}'$ (i.e. the RSR estimator). This is particular means that if one can show that the right hand side above is small for the true density $v$, then the solution $u$ must be close to $v$. Thus we can concentrate on analyzing the simpler expression $\|\nabla L(v)\|_{\mathcal{H}^a}$ rather than working with the optimizer $u$ directly. Remarkably, this result is a pure Hilbert space result, and it holds for any kernel.

We now thus turn to the task of bounding $\|\nabla L(v)\|_{\mathcal{H}^a}$. As discussed in Section 3.2, the gradient of $L$ in $\mathcal{H}^a$ is given by

$$\nabla L(f) = -\frac{1}{N}\sum f(x_i)^{-1}k_{x_i} + f. \tag{106}$$

Opening the brackets in $\|\nabla L(v)\|^2_{\mathcal{H}^a}$ we have

$$\|\nabla L(v)\|^2_a = \frac{1}{N^2} \sum_{i,j} v^{-1}(x_i) v^{-1}(x_j) k(x_i, x_j) - 2\frac{1}{N} \sum_i v^{-1}(x_i) \langle k_{x_i}, v \rangle_a + \|v\|^2_a \quad (107)$$

$$= \frac{1}{N^2} \sum_{i,j} v^{-1}(x_i) v^{-1}(x_j) k(x_i, x_j) - 2 + \|v\|^2_a. \quad (108)$$

By definitions, we clearly have $\|v\|^2_a - 1 \to 0$ when $a \to 0$. Thus we have to show that the first term in (108) concentrates around 1. To this end, we will first show that the expectation of

$$\frac{1}{N^2} \sum_{i,j} v^{-1}(x_i) v^{-1}(x_j) k(x_i, x_j) \quad (109)$$

(with respect to $x_i$'s) converges to 1. This is really the heart of the proof, as here we show why $v$ approximately minimizes the RSR objective, in expectation, by exploiting the interplay between the kernel, the regularizing coefficient, and the density $v$. This argument is carried out in Lemmas 13, 15, and 16.

Once the expectation is understood, we simply use the Chebyshev inequality to control the deviation of (109) around its mean. This requires the control of cross products of the terms in (109) and can be achieved by arguments similar to those used to control the expectation. This analysis is carried out in Propositions 17 - 19, and Lemma 20.

One of the technically subtle issues throughout the proof is the presence of the terms $v^{-1}(x_i)$, due to which some higher moments and expectations are infinite. This prevents the use of standard concentration results and requires careful analysis. This is also the reason why obtaining convergence rates and high probability bounds is difficult, although we believe this is possible.

## L.2 Full Proof Details

Throughout this section let $L_2$ be the $L_2$ space of the the Lebesgue measure on $\mathbb{R}^d$, $L_2 = \left\{ f : \mathbb{R}^d \to \mathbb{C} \mid \int_{\mathbb{R}^d} |f|^2 \, dx \leq \infty \right\}$.

Next, we observe that $L(f)$ is *strongly* convex with respect to the norm $\|\cdot\|_{\mathcal{H}^a}$ (see Nesterov (2003) for an introduction to strong convexity). As a consequence, we have the following:

**Lemma 12.** *Let $u$ be the minimizer of $L$ in $\mathcal{C}'$. Then for every $v \in \mathcal{C}'$,*

$$\|u - v\|_{\mathcal{H}^a} \leq \|\nabla L(v)\|_{\mathcal{H}^a}. \quad (110)$$

*Proof.* The function $f \mapsto \|f\|^2_{\mathcal{H}^a}$ is 2-strongly convex with respect to $\|\cdot\|^2_{\mathcal{H}^a}$. Since $L(f)$ is obtained by adding a convex function, and multiplying by $\frac{1}{2}$, it follows that $L(f)$ is 1-strongly convex. Strong convexity implies that for all $u, v \in \mathcal{C}'$ we have

$$\langle \nabla L(v) - \nabla L(u), v - u \rangle_a \geq \|v - u\|^2_a, \quad (111)$$

see Nesterov (2003), Theorem 2.1.9. Using the Cauchy Schwartz inequality and the fact that $u$ is a local minimum with $\nabla L(u) = 0$, we obtain

$$\|\nabla L(v)\|_a \cdot \|u - v\|_a \geq \langle \nabla L(v), v - u \rangle_a \geq \|v - u\|^2_a, \quad (112)$$

yielding the result. $\qquad \square$

Now, suppose the samples $x_i$ are generated from a true density $(f^*)^2$. Let $v := f^*$ be the square root of this density. Then, to show that the estimator $u$ is close to $v$, it is sufficient to show that $\|\nabla L(v)\|_a$ is small. We write $\nabla L(v)$ explicitly:

$$\|\nabla L(v)\|^2_a = \frac{1}{N^2} \sum_{i,j} v^{-1}(x_i) v^{-1}(x_j) k(x_i, x_j) - 2\frac{1}{N} \sum_i v^{-1}(x_i) \langle k_{x_i}, v \rangle_a + \|v\|^2_a \quad (113)$$

$$= \frac{1}{N^2} \sum_{i,j} v^{-1}(x_i) v^{-1}(x_j) k(x_i, x_j) - 2 + \|v\|^2_a. \quad (114)$$

Since $v$ is a fixed function in $\mathcal{H}^1$, it is clear from definition (8) that $\|v\|_a^2 - 1 \to 0$ as $a \to 0$. Thus, to bound $\|\nabla L(v)\|_a^2$, it suffices to bound

$$\frac{1}{N^2} \sum_{i,j} v^{-1}(x_i) v^{-1}(x_j) k(x_i, x_j) - 1 \tag{115}$$

with high probability over $x_i$, when $N$ is large and $a$ is small.

Note that the form (10) of the kernel $k^a$ implies that this is a stationary kernel, i.e. $k^a(x,y) = g^a(x-y)$ with

$$g^a(x) = \int_{\mathbb{R}^d} \frac{e^{2\pi i \langle -x, z \rangle}}{1 + a \cdot (2\pi)^{2m} \|z\|^{2m}} dz. \tag{116}$$

**Lemma 13.** *Let $k^a(x,y)$ be the SDO kernel defined by (10). For any function $v \in L_2$, and $x \in \mathbb{R}^d$ set*

$$(K_a v)(x) = \int k^a(x,y) v(y) dy. \tag{117}$$

*Then $K_a$ is a bounded operator from $L_2$ to $L_2$, with $\|K_a\|_{op} \leq 1$ for every $a > 0$. Moreover, for every $v \in L_2$, $\|K_a v - v\|_{L_2} \to 0$ with $a \to 0$.*

*Proof.* Note first that $K_a$, given by (117), is a convolution operator, i.e. $K_a v = g^a * v = \int g^a(x-y) v(y) dy$. Further, by the Fourier inversion formula, the Fourier transform of $g^a$ satisfies $\mathcal{F} g^a(z) = \frac{1}{1 + a \cdot (2\pi)^{2m} \|z\|^{2m}}$ (see Section D.1 for further details), and recall that $\mathcal{F}(g^a * v) = \mathcal{F} g^a \cdot \mathcal{F} v$. Since $|\mathcal{F} g^a(z)| \leq 1$ for every $z$ and $a > 0$, this implies in particular that $K_a$ has operator norm at most 1. Next, by the Plancharel equality we have

$$\|K_a v - v\|_{L_2}^2 = \|\mathcal{F}(K_a v - v)\|_{L_2}^2 \tag{118}$$

$$= \left\| \mathcal{F}(v) \cdot \left( \frac{1}{1 + a \cdot (2\pi)^{2m} \|z\|^{2m}} - 1 \right) \right\|_{L_2}^2 \tag{119}$$

$$= \left\| \mathcal{F}(v) \cdot \left( \frac{a \cdot (2\pi)^{2m} \|z\|^{2m}}{1 + a \cdot (2\pi)^{2m} \|z\|^{2m}} \right) \right\|_{L_2}^2 \tag{120}$$

$$= \int |\mathcal{F}(v)(z)|^2 \cdot \left( \frac{a \cdot (2\pi)^{2m} \|z\|^{2m}}{1 + a \cdot (2\pi)^{2m} \|z\|^{2m}} \right)^2 dz \tag{121}$$

Fix $\varepsilon > 0$. For a radius $r$ denote by $B(r) = \{z \mid \|z\| \leq r\}$ the ball of radius $r$, and let $B^c(r)$ be its complement. Since $\int |\mathcal{F}(v)(z)|^2 dz < \infty$, there is $r > 0$ large enough such that $\int_{B^c(r)} |\mathcal{F}(v)(z)|^2 dz \leq \varepsilon$. We bound (121) on $B(r)$ and $B^c(r)$ separately. Choose $a > 0$ such that $a \leq \left((2\pi)^{2m} r^{2m}\right)^{-1} \cdot \varepsilon^{\frac{1}{2}} \cdot \|v\|_{L_2}^{-1}$. Then

$$\int |\mathcal{F}(v)(z)|^2 \cdot \left( \frac{a \cdot (2\pi)^{2m} \|z\|^{2m}}{1 + a \cdot (2\pi)^{2m} \|z\|^{2m}} \right)^2 dz \tag{122}$$

$$= \int_{B(r)} |\mathcal{F}(v)(z)|^2 \cdot \left( \frac{a \cdot (2\pi)^{2m} \|z\|^{2m}}{1 + a \cdot (2\pi)^{2m} \|z\|^{2m}} \right)^2 dz + \int_{B^c(r)} |\mathcal{F}(v)(z)|^2 \cdot \left( \frac{a \cdot (2\pi)^{2m} \|z\|^{2m}}{1 + a \cdot (2\pi)^{2m} \|z\|^{2m}} \right)^2 dz \tag{123}$$

$$\leq \varepsilon \cdot \|v\|_{L_2}^{-1} \int_{B(r)} |\mathcal{F}(v)(z)|^2 dz + \int_{B^c(r)} |\mathcal{F}(v)(z)|^2 dz \tag{124}$$

$$\leq \varepsilon + \varepsilon. \tag{125}$$

This completes the proof. $\qquad\square$

**Assumption 14.** *Assume that the density $v^2$ is compactly supported in $\mathbb{R}^d$, and denote the support by $B = supp(v^2)$.*

Note in particular that this implies that $B$ is of finite Lebesgue measure, $\lambda(B) \leq \infty$.

We treat the components with $i \neq j$ and $i = j$ in the sum (115) separately. In the case $i = j$ set $Y_i = v^{-2}(x_i)k(x_i, x_i) = g_a(0)v^{-2}(x_i)$, where $g^a$ was defined in (116). Note that the variable $Y_i$ has an expectation, but does not necessarily have higher moments. Nevertheless, it is still possible to bound the sum using the Marcinkiewicz-Kolmogorov strong law of large numbers, see Loève (1977), Section 17.4, point $4°$. We record it in the following Lemma, where we also allow $a$ to depend on $N$, denoting $a = a(N)$, and assume that $a(N)$ does not decay too fast with $N$.

**Lemma 15.** *Assume that* $\lim_{N \to \infty} a^{-\frac{d}{2m}}(N)/N = 0$. *Then*

$$\frac{g^{a(N)}(0)}{N^2} \sum_i v^{-2}(x_i) \to 0 \tag{126}$$

*almost surely, with* $N \to \infty$.

Note that since $2m > d$ by construction of the kernels, $a(N) = 1/N$ satisfies the above decay assumption.

*Proof.* We have

$$\mathbb{E}v^{-2}(x) = \int v^{-2}(x) \cdot v^2(x)\mathbb{1}_{\{B\}}(x)dx = \lambda(B) < \infty. \tag{127}$$

Thus the Marcinkiewicz-Kolmogorov law implies that

$$\frac{1}{N} \sum_i v^{-2}(x_i) \to \lambda(B) \tag{128}$$

almost surely. Next, recall that by Proposition 7, we have $g^a(0) = a^{-\frac{d}{2m}}g^1(0)$. Our decay assumption on $a(N)$ implies then that $g^{a(N)}(0)/N \to 0$, which together with (128) completes the proof. $\square$

Next, for $i \neq j$, set $Y_{ij} = v^{-1}(x_i)v^{-1}(x_j)k(x_i, x_j)$.

**Lemma 16.** *For every* $a > 0$ *we have* $|\mathbb{E}Y_{ij}| \leq 1$, *and moreover* $\mathbb{E}Y_{ij} \to 1$ *when* $a \to 0$.

*Proof.* Recall that $v^2$ is a density, i.e. $\int v^2(x)dx = 1$, and that the operator $K^a$ was defined in (117). We have

$$|\mathbb{E}Y_{ij} - 1| = \left| \int v^{-1}(x)v^{-1}(y)k^a(x, y)v^2(x)v^2(y)dxdy - \int v^2(x)dx \right| \tag{129}$$

$$= \left| \int k^a(x, y)v(x)v(y)dxdy - \int v^2(x)dx \right| \tag{130}$$

$$= \left| \int dx \cdot v(x)\left[(K_a v)(x) - v(x)\right] \right| \tag{131}$$

$$\leq \|v\|_{L_2} \|(K_a v) - v\|_{L_2} \tag{132}$$

$$= \|(K_a v) - v\|_{L_2}. \tag{133}$$

The second statement now follows from Lemma 13. For the first statement, write

$$|\mathbb{E}Y_{ij}| = \langle K_a v, v \rangle_{L_2} \leq \|K_a v\|_{L_2} \|v\|_{L_2} \leq 1, \tag{134}$$

where we have used the Cauchy-Schwartz inequality and the first part of Lemma 13. $\square$

It thus remains to establish that $\frac{1}{N^2} \sum_{i \neq j}(Y_{ij} - EY_{ij})$ converges to 0 in probability. We will show this by bounding the second moment,

$$\frac{1}{N^4}\mathbb{E}\left( \sum_{i \neq j}(Y_{ij} - EY_{ij}) \right)^2 = \frac{1}{N^4} \sum_{i \neq j, i' \neq j'} (\mathbb{E}Y_{ij}Y_{i'j'} - \mathbb{E}Y_{ij}\mathbb{E}Y_{i'j'}) \tag{135}$$

and then using the Chebyshev inequality. Observe that there are three types of terms of the form $(\mathbb{E}Y_{ij}Y_{i'j'} - \mathbb{E}Y_{ij}\mathbb{E}Y_{i'j'})$ on the right hand side of (135). The first type is when $\{i, j\} = \{i', j'\}$ as sets. Second is when $|\{i, j\} \cap \{i', j'\}| = 1$, and the third is when $\{i, j\}$ and $\{i', j'\}$ are disjoint. In the following three propositions, we bound each type of terms separately.

**Proposition 17.** *There is a function of $m$, $c(m) > 0$, such that*

$$\mathbb{E}Y_{ij}^2 \leq \lambda(B)a^{-\frac{d}{2m}}c(m) < \infty. \tag{136}$$

*Proof.* Write

$$\mathbb{E}Y_{ij}^2 = \int v^{-2}(x)v^{-2}(y)(k^a(x,y))^2 v^2(x)v^2(y)dxdy \tag{137}$$

$$= \int \mathbb{1}_{\{B\}}(x)\mathbb{1}_{\{B\}}(y)(k^a(x,y))^2 dxdy \tag{138}$$

$$\leq \int \mathbb{1}_{\{B\}}(x) \|k_x^a\|_{L_2}^2 dx \tag{139}$$

$$= \lambda(B)a^{-\frac{d}{2m}}c(m), \tag{140}$$

where we have used Lemma 8 to compute $\|k_x^a\|_{L_2}^2$. $\qquad\square$

For the $|\{i,j\} \cap \{i',j'\}| = 1$ case we have

**Proposition 18.** *Let $i, j, t$ be three distinct indices. Then*

$$|\mathbb{E}Y_{ij}Y_{jt}| \leq 1. \tag{141}$$

*Proof.*

$$\mathbb{E}Y_{ij}Y_{jt} = \int v^{-1}(x_i)v^{-1}(x_j)k^a(x_i,x_j)v^{-1}(x_j)v^{-1}(x_t)k^a(x_j,x_t)v^2(x_i)v^2(x_j)v^2(x_t)dx_idx_jdx_t \tag{142}$$

$$= \int v(x_i)k^a(x_i,x_j)v(x_t)k^a(x_t,x_j)\mathbb{1}_{\{B\}}(x_j)dx_idx_jdx_t \tag{143}$$

$$= \int (K_a v)^2(x_j)\mathbb{1}_{\{B\}}(x_j)dx_j \tag{144}$$

$$\leq \int (K_a v)^2(x_j)dx_j \tag{145}$$

$$= \|K_a v\|_{L_2}^2 \tag{146}$$

$$\leq 1, \tag{147}$$

where we have used Lemma 13 on the last line. $\qquad\square$

Finally, for the disjoint case,

**Proposition 19.** *Let $i, j, i', j'$ be four distinct indices. Then*

$$\mathbb{E}(Y_{ij} - EY_{ij})(Y_{i'j'} - EY_{i'j'}) = 0. \tag{148}$$

*Proof.* When $i, j, i', j'$ are distinct, $Y_{ij}$ and $Y_{i'j'}$ are independent. $\qquad\square$

We now collect these results to bound (135).

**Lemma 20.** *There is a function $c'(m, B) > 0$ of $m$, $B$, such that, choosing $a(N) = 1/N$, we have*

$$\frac{1}{N^4}\mathbb{E}\left(\sum_{i \neq j}(Y_{ij} - EY_{ij})\right)^2 \leq \frac{c'(m,B)}{N} \tag{149}$$

*for every $N > 0$.*

*Proof.* Observe first that all the expressions of the form $\mathbb{E}Y_{ij}\mathbb{E}Y_{i'j'}$ are bounded by 1, by Lemma 16. Next, note that there are $O(N^2)$ terms of the first type. Since $2m > d$, we have $a^{-\frac{d}{2m}} = a^{-\frac{d}{2m}}(N) \leq N$, and thus, the overall contribution of such terms to the sum in (149) is $O(N^{-4} \cdot N \cdot N^2) = O(1/N)$. Similarly, there are $O(N^3)$ terms of the second type, each bounded by constant, and thus the overall contribution is $O(N^{-4} \cdot N^3) = O(1/N)$. And finally, the contribution of the terms of the third type is 0. $\qquad\square$

We now prove the main consistency Theorem.

*Of Theorem 11.* First, observe that by definition $\|u - v\|_{L_2} \leq \|u - v\|_a$ for any $u, v \in \mathcal{H}^1$. Next, by Lemma 12 and using (114), we have

$$\|u_N - v\|_{L_2} \leq \|u_N - v\|_a \leq \frac{1}{N^2} \sum_{i,j} v^{-1}(x_i) v^{-1}(x_j) k(x_i, x_j) - 2 + \|v\|_a^2. \qquad (150)$$

Clearly, by definition (8), we have $\|v\|_a^2 - 1 \to 0$ as $a \to 0$. Write

$$\frac{1}{N^2} \sum_{i,j} v^{-1}(x_i) v^{-1}(x_j) k(x_i, x_j) - 1 = \frac{1}{N^2} \sum_i Y_i + \frac{1}{N^2} \sum_{i \neq j} (Y_{ij} - \mathbb{E} Y_{ij}) + \left[ \frac{N^2 - N}{N^2} \cdot \mathbb{E} Y_{12} - 1 \right].$$
$$(151)$$

The last term on the right hand side converges to $0$ deterministically with $a$ and $N$, by Lemma 16. The first terms converges strongly, and hence in probability, to $0$, by Lemma 15. And finally, the middle term converges in probability to $0$ by Lemma 20 and by Chebyshev inequality. □

## M COMPARISON OF RSR FOR DIFFERENT KERNELS

In this section, we evaluate the RSR density estimator with various kernels on the ADBench task. Specifically, we consider the SDO kernel given by (10), the $L_2$ Laplacian kernel, and the Gaussain kernel. The Laplacian (aka Exponential) kernel is given by

$$k(x, y) = (1/\sigma^d) \cdot e^{-\|x-y\|/\sigma} \tag{152}$$

for $\sigma > 0$, where $\|x - y\|$ is the Euclidean norm on $\mathbb{R}^d$. The Gaussian kernel is given by

$$k(x, y) = (1/\sigma^d) \cdot e^{-\|x-y\|^2/(2\sigma^2)}. \tag{153}$$

Generally, on this task observe very similar behavior for all the kernels.

Figure 5 presents the performance of each kernel on the ADBench benchmark (see section 6.1). Specifically, we plot the test set AUC values on the Y-axis against the different datasets on the X-axis. Note that due to convexity, the results are nearly identical for different initializations. The vertical lines around the each dot represent the standard deviation, while the dots represent the mean over multiple runs of the same experiment. Most of the randomness comes from the random sampling using on computing the Fisher Divergeneces, which are used for selecting the hyperparametrs $(a,\sigma)$.

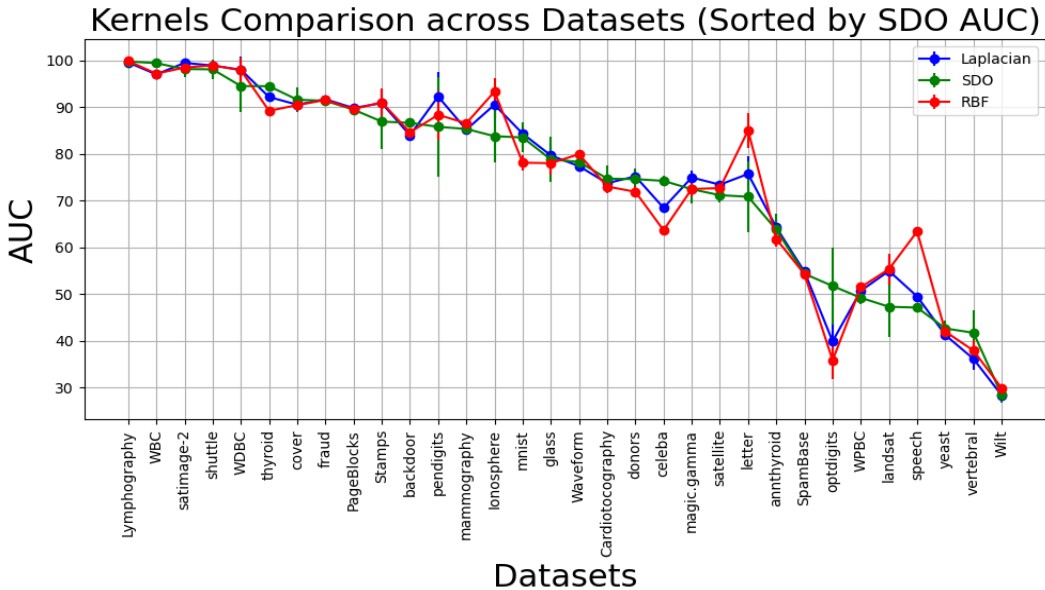

Figure 5: Comparing RSR results on ADBench for different kernel.

Due to the time constraints of the review discussion, a subsampling of 1500 points was employed for each dataset. In addition, on one dataset, the 'lymphography' dataset, the FD-based hyper-parameter tuning produced an error, and on this dataset (and only on this dataset) we have manually chosen the hyperparameters by inspecting the FD curve.

