# OpenReview forum: "Unnormalized Density Estimation with Root Sobolev Norm Regularization"
_ICLR.cc/2024/Conference — Submitted to ICLR 2024_

### Official Review · Reviewer_CL4E · 2023-10-24

**Soundness:** 3 good
**Presentation:** 2 fair
**Contribution:** 2 fair
**Rating:** 3
**Confidence:** 3

**Summary:**

This paper studies the density estimation using the likelihood with the Sobolev norm regularization. Unnormalized models are employed for non-parametric estimation. The algorithm is implemented using gradient-based learning. The authors proposed sampling approximation for computing the RKHS norm. Then, they investigate the difference between the proposed method (RSR) and the standard kernel density estimator (KDE). Indeed, two estimators provide different results for the separability of the cluster structure. Some numerical experiments indicate that the proposed method outperforms most of the existing works for anomaly detection. Also, computational properties are numerically analyzed.

**Strengths:**

- The formulation of the proposed method (RSR) is simple and easy to understand.
- The authors studied some properties of the proposed estimator, e.g., the ratio of the estimated probability densities shown in Section 5.

**Weaknesses:**

- The proposed estimator is rather straightforward and has less impact the machine learning community.
- Once the estimator for the unnormalized model is estimated, how can the estimator be utilized?
- Section 6.3 reports the ratio of negative values for the estimated function f. I'm not sure why the ratio of negative values is important to assess the computational properties.
- Sobolev space is closely related to the RKHS with Matern kernel; see [1] below. Some supplementary comments on that relationship would be helpful for researchers interested in theoretical analysis of the learning algorithms.

[1] Gregory E. Fasshauer and Qi Ye, Reproducing kernels of generalized Sobolev spaces via a Green function approach with distributional operators, arXiv:1204.6448.

**Questions:**

- The estimator is similar to the one proposed in Ferraccioli's JRSS paper. Please clarify the main difference between them.
- Once the estimator for the unnormalized model is estimated, how can the estimator be utilized? Showing an example of using an unnormalized model would be helpful for readers.
- Section 6.3 reports the ratio of negative values for the estimated function f. I'm not sure the reason why the ratio of negative values is important to assess the computational properties.
- Sobolev space is closely related to the RKHS with Matern kernel; see [1] below. Some supplementary comments would be helpful for researchers who are interested in theoretical analysis of the learning algorithms.
[1] Gregory E. Fasshauer and Qi Ye, Reproducing kernels of generalized Sobolev spaces via a Green function approach with distributional operators, arXiv:1204.6448.

---

> ### Author Response · Authors · 2023-11-16
>
> We thank the reviewer for the feedback.
>
>
>
> >Sobolev space is closely related to the RKHS with Matern kernel; see [1] below. Some supplementary comments on that relationship would be helpful for researchers interested in theoretical analysis of the learning algorithms.
>
> Thank you for including the reference [1]. It contains a clear relevant results on the Matern kernel in context of Sobolev space and we will include it in the paper.
> Please see the discussion in the global "General Reply" comment on the relation between the SDO kernel and Matern kernels.
>
>
> >The estimator is similar to the one proposed in Ferraccioli's JRSS paper. Please clarify the main difference between them.
>
> We discuss the relation with the paper
>
> *Nonparametric density estimation over complicated domains, Ferraccioli et al,  JRSS 2020*
>
> In the above paper, the authors also use a reparametrisation to ensure non-negativity of the density, and regularise using  a version of a Sobolev space. However, as we discuss in our introduction and literature review sections,
> that general principle is known since the 1980s for one dimension. The challenge is how one can implement the principle in $d>1$.
>
>
> The above mentioned paper *is solely restricted to two dimensions, d=2*. (both for applications and theory). In particular, to obtain their solutions, they discretize (triangulate) the domain of interest in $R^2$.
> This is clearly infeasible in any higher dimension.
>
>
> In contrast, the approach we develop, including the natural gradient optimisation and the use of Fisher divergence for parameter tuning, allow us to obtain good anomaly detection performance on datasets with $d>100$.
> Moreover, we consider the fact that such methods *are* useful in high dimensional situations is a significant contribution by itself, since such methods were not considered in recent literature in high dimensions. We will emphasize this more in the discussion in the paper.
> Would the reviewer agree that this is indeed a contribution of interest?
>
>
> > Section 6.3 reports the ratio of negative values for the estimated function f. I'm not sure why the ratio of negative values is important to assess the computational properties.
>
> This point is important and we now clarify it.
> As we discuss in the paragraph following Eq. (5), the main optimisation objective (5) is not convex as a function of $f$. Moreover, if one nevertheless tries to optimise (5) in $f$ (i.e., in $\alpha$, using $f_{\alpha}$) directly, one would obtain very poor results. The gradient descent will get stuck in a local optimum with poor values.
> We can provide an experimental demonstration of this if that would be of interest.
>
>
> However, we observe that (5) is convex on the non-negative cone,
> i.e., $f$ s.t. $f>0$ everywhere. (It is also sufficient to require  only $f(x_i)>0$ on data points $x_i$).
> Thus, if one restricts to such cone, one can obtain the optimal solution on that cone. We observe that such solutions are empirically much better.
> Next, we note that solving a constrained optimisation problem, requiring the solutions to be restricted to the cone is computationally difficult.  Moreover, since the cone is not closed (topologically), this is also not quite clear theoretically, since projection methods can not be applied.
>
>
> To resolve this, we observe that  *natural gradient* preserves the cone.
> This means that one can in fact do *unconstrained* optimisation.
> This is true for non-negative kernels, such as the Matern kernels (discussion following Equation (5) and Proposition 10), and is approximately true for the SDO kernel.
>
> In the experiment in Section 6.3 we show that empirically, for the SDO kernel, natural gradient indeed tends to preserve positivity, while the regular gradient does not. This demonstrates that natural gradient is indeed better for optimisation purposes. Note that, theoretically, this is one of the very few settings were we know *why* the natural gradient is better (i.e. due to positivity).
>
> Moreover, we can also directly show that the results in Fig 2, our main empirical results, would be worse if regular gradient descent is used instead of natural.
>
> Does this answer make sense? We would be glad to provide additional clarifications.
>
>
>
>
> Continued in the comment below.

---

> > ### Author Response · Authors · 2023-11-16
> >
> > >Once the estimator for the unnormalized model is estimated, how can the estimator be utilized?
> >
> > We are not sure we understand this question. We have included an application
> > of the method to a state of the art anomaly detection problem, and showed that the method performs very well, outperforming many dedicated anomaly detection methods on many datasets.  In addition, in the future we hope to develop  Markov Chain Monte Carlo methods of sampling the density, thus turning the model into a generative model. Metropolis sampling algorithms do not require the density to be normalised, and thus are well suited for such situations.
> >
> > Does this resolve the question of utilisation?
> >
> >
> >
> > >The proposed estimator is rather straightforward and has less impact the machine learning community.
> >
> > We will make clarifications in the text, re-emphsizing and incorporating many of the points in the comments above. In view of this, would the reviewer consider modifying the above statement?
> >
> > We further note, that as the reviewer pointed out,
> > a *two-dimensional* case was published only three years ago in JRSS, while here we treat high dimensions, and also obtain very competitive results on a  mainstream AD benchmark.  We believe this may put the impact of the paper in a different perspective.
> >
> > In connection with straightforwardness of the methods, we believe this would ultimately always remain somewhat hard to argue about. However, as we mention above, there are many components beyond the formulation itself that are required to make the approach practical, and we feel that it is not obvious how that could be done or even whether it could be done at all.

---

> ### Author Response · Authors · 2023-11-21
>
> Dear Reviewer CL4E,
> Thank you again for your feedback. In our earlier comments we have
> answered the questions posed in the review and addressed the concerns raised in the Weaknesses section.
>
> Have all the concerns been addressed, or are there any issues still remaining?

---

### Official Review · Reviewer_o1rt · 2023-10-29

**Soundness:** 4 excellent
**Presentation:** 3 good
**Contribution:** 2 fair
**Rating:** 5
**Confidence:** 3

**Summary:**

This paper proposes a novel framework for un-normalized kernelized density estimation that differs from common practice mainly in 2 aspects:

- Kernel density estimation generally uses $ \hat{f} = \sum_{i=1}^N \alpha_i k_{x_i} $ with a non-negative coefficients $\alpha_i$ and non-negative valued-kernel $k$; the proposes $\tilde{f} = \left(\sum_{i=1}^N \beta_i k_{x_i}\right)^2$ with un-constrained $\beta_i$ and $k$, which trades away normalization with a different objective in regularization. The paper presents contrived example about the difference between the two density estimators and argue in favor of using the squared density estimator for spectral clustering and anomaly detection.
- The paper proposes the SDO kernel, a stationary kernel whose RKHS norm involves derivatives of $f$ to a certain order. The paper notes that the kernel has a tractable spectral density (up to a normalization constant), and proposes a random Fourier feature approximation based on MCMC sampling of the spectral density.

**Strengths:**

- Soundness: the paper establishes a sound theoretical framework for un-normalized density estimation.
- Novelty: (i) the authors propose a novel SDO kernel, which furthers our understanding on the relation between Sobolev spaces and the spectral densities of kernels; (ii) the argument how RSR differs from KDE is interesting and presents a good motivation to use this framework for density estimation.
- Experiments: the paper mainly focuses on anomaly detection with overall good empirical results.

**Weaknesses:**

- Motivation: I admit my lack of familiarity in un-normalized density estimation but I hope the authors can help clarify on a few issues: (i) I think the paper makes a sound argument about using the squared version of a linear combination of kernels by comparing it against KDE, but the decision to use a _single_ derivative order lacks motivation in my view. I pose this question in the next section, and I believe that a well-written section on this topic should be included in the paper. (ii) the paper mentions on the top section of p. 2 that regularizing over $ |f^*|_{L_2} $ is a desirable property: what does it mean?
- Presentation: the paper is well-written overall, but the latter parts of the manuscript seems in a draft form: (i) the manuscript lacks a final summary section, and the Fisher divergence and hyperparameter tuning section seems better placed in an early section when the authors present the methodology; (ii) The paper mentions the consistency of the estimator in the main manuscript, but delays all discussions about consistency to the supplementary materials - I believe that it would be helpful to bring up the main theorems for consistency in the main text.

**Questions:**

I have one main question about the objective of the SDO kernel: the paper proposes an RKHS norm that involves the L2 norm of the function, and also its derivatives _at_ a certain degree $m$, leading to a kernel with no analytical expression, but has a tractable spectral density (up to a normalization constant). The type of Sobolev norm $ \sum_{|\kappa|_1=m} ||D^{\kappa} f||^2 $ seems quite unusual.

We know that the original Sobolev norm involves derivatives _up_ to degree $m$: $ \sum_{|\kappa|_1\leq m} ||D^{\kappa} f||^2 $, and the Sobolev spaces are norm-equivalent to the RKHS of a Matérn kernel (in closed form and well-studied). Could the authors explain why their choice of the Sobolev norm is useful, and what will the results be like if one uses the Matérn kernel in replacement of the SDO kernel? I think a good justification for this model choice is a useful addition to the paper, as the Matérn kernel is sufficiently close to a regularization on Sobolev norm, and the SDO kernel seems more difficult because of the reliance on using Fourier features to approximate its kernel values.

- The paper's main method is marked as "INER" in Figures 1 and 2, but the text makes no mention of what it stands for: is this a mistake?

---

> ### Author Response · Authors · 2023-11-16
>
> Thank you for the feedback!
>
> We agree with the point raised in this review that the specific choice of the SDO kernel requires more motivation. Please see a detailed discussion of our motivation, and of relations of SDO with other kernels, in the General Reply comment common to all reviews. In addition, in a few days we will include experiments that evaluate our algorithms with the Laplacian and Gaussian kernels, which correspond to a representative range of Sobolev related kernels and have analytic expressions. As we mention in the above discussion, all contributions of the paper besides the SDO kernel are relevant for these kernels too.
>
>
> We comment on an additional point on equivalence of Sobolev spaces, which was mentioned in the review.  It is true for instance that the spaces corresponding to the Laplacian and SDO kernels are equivalent for all values of the Laplacian parameter $\sigma$ and SDO parameter $a$ (see the discussion above). However, this only means that the spaces *contain* the same set of functions. This in turn implies that the norms are equivalent, but the equivalence constants diverge with the dimension. Since for regularization we are interested in an actual value of the norm, the resulting algorithms will be different between such spaces. Consider for instance that $\|\cdot \|_1$ and $\|\cdot\|_2$ norms on $R^d$ are equivalent norms, but with drastically different regularisation properties. Thus just norm equivalence is too coarse a notion to distinguish (or unify) between different regularisation algorithms.
>
>
> Do these comments address the issue with the SDO choice?
>
>
> > (ii) the paper mentions on the top section of p. 2 that regularizing over  $\|f\|_{L_2}^2$ is a desirable property: what does it mean?
>
> Our intention here was to say by using the Sobolev type kernel, we automatically deal with functions that satisfy $\inf f^2(x)dx < \infty$. Such functions are densities up to a constant. Non integrable functions on the other hand can not represent densities (for the purposes of sampling from them, for instance). This is not a trivial condition, as there is no reason why an "arbitrary" function should be integrable.
>
>
>
> >The paper's main method is marked as "INER" in Figures 1 and 2, but the text makes no mention of what it stands for: is this a mistake?
>
> Yes, it should be RSR. Will be fixed.
>
>
> >The paper mentions the consistency of the estimator in the main manuscript, but delays all discussions about consistency to the supplementary materials - I believe that it would be helpful to bring up the main theorems for consistency in the main text.
>
> We will definitely consider this. The challenge here is that there seems to be no other part of the paper which is non crucial enough to be moved to the supplementary instead.

---

> > ### Author Response · Authors · 2023-11-21
> >
> > Dear Reviewer o1rt,
> > Thank you again for your feedback. In our earlier comments we have
> > answered the questions posed in the review and addressed the concerns raised in the Weaknesses section.
> >
> > Have all the concerns been addressed, or are there any issues still remaining?

---

### Official Review · Reviewer_oZU9 · 2023-11-02

**Soundness:** 3 good
**Presentation:** 3 good
**Contribution:** 2 fair
**Rating:** 5
**Confidence:** 3

**Summary:**

This paper proposes a new nonparametric density estimator based on regularized maximum likelihood estimation. The estimate is represented as the square of its square root, ensuring non-negativity at the cost of not having unit mass, which the paper argues is sufficient for applications, such as anomaly detection or sampling, that can utilize such an unnormalized density estimate.

**Strengths:**

The proposed method is fairly clearly well-explained. The empirical results on the anomaly detection benchmark are quite impressive and show the proposed method is useful in a number of real-world problems.

**Weaknesses:**

**Major**

1. Page 1, Paragraph 1:
> *While there is recent work for low dimensional (one or two dimensional) data... there still are very few non-parametric methods applicable in higher dimensions.*

I'm not convinced by this motivation, for a two reasons. First, many nonparametric methods have been described for data of arbitrary dimension (including in some of the cited papers). However, standard minimax lower bounds show that high-dimensional non-parametric density estimation is statistically intractable, in terms of many performance metrics. Thus, I'm not convinced that a new method will perform well in high dimensions unless it makes some more explicit assumptions on the density being estimated. Second, while recent neural network based density estimation methods are technically parametric, the complexity of these models is so large that they often behave more like non-parametric methods; i.e., they can fit quite complex densities. So it's not clear that a new nonparametric method should outperform these models in practical applications.

2. Eq. (1): This objective is unbounded if the order of the derivative $D$ is $\leq d/2$ (since the Sobolev $W^{k,2}(R^d)$ contains singularities, which can be centered on the samples.) The assumption $m > d/2$ in Theorem 2 addresses this, but, when first seeing Eq. (1), I was a bit confused by this. So perhaps it is worth mentioning this assumption earlier.

3. Section 1, Last Paragraph, and Section 2, Last Paragraph: Both of these briefly mention consistency, but no details are provided here regarding the type of consistency (in $L_2$ in probability) or the assumptions made.

4. Section 3.1, just after Eq. (5)
> "there seems to be no computationally affordable way to restrict the optimization to the positive cone $\mathcal{C}$... We resolve this issue in two steps..."

Although it seems reasonable in practice, the solution proposed here is *ad hoc*, and it's not clear how this relates to the consistency guarantee (Theorem 11). Ultimately, in Appendix K, the paper assumes "positivity on the data points $x_i$", and it's not clear whether this condition is likely to hold as $n \to \infty$. On one hand, the estimate might approach the true (non-negative) density, but, on the other hand, the number of points $x_i$ is increasing. So, I think this is a big hole in the consistency guarantee. To fix this, the paper should analyze whether the "positivity on the data points $x_i$" condition holds as $n \to \infty$, and whether any additional conditions are necessary to ensure this (e.g., it might suffice if the true density is bounded away from $0$?).

5. While the experiments demonstrate impressive performance on an anomaly detection benchmark, it's not clear from the paper where this advantage comes from or whether it is statistically significant or simply due to chance. One way to strengthen this would be an experiment on synthetic data where one can clearly (i.e., in an intuitive and unambiguous manner) see the advantage of RSR. The synthetic experiment in Section 5 almost does this, but it doesn't go as far as quantifying the advantage of RSR.

6. I didn't really understand the purpose of Section 5. The two curves in Figure 2(b) seem almost identical, up to a constant scaling factor. Since the $y$-axis log likelihood, this just looks like INER=(KDE)$^2$, up to a constant factor. Related to this, I didn't understand why "the gap between the values on the first and second cluster [being] larger for the RSR model" explains why RSR is better for anomaly detection -- what matters for anomaly detection is probably the *ratio of the between-cluster variation to the within-cluster variation*, but the within-cluster variation is also much bigger for RSR (INER) than for KDE.

**Minor**

1. Usually $\mathcal{H}^a$ denotes the Sobolev space $W^{a, 2}$ (i.e., the space of $L_2$ functions with $a^{th}$ derivatives in $L_2$). I suggest aligning the use of $\mathcal{H}^a$ in this paper with this more standard notation. In particular, it is strange to me that the derivative order is not explicitly denoted in this paper's usage of $\mathcal{H}^a$.

2. Page 2, Paragraph 1, Last Sentence: Should "|f^*|_{L_2}$" be "||f^*||_{L_2}$"? I believe this is a proper norm...

3. Section 4, "Single Derivative Order kernel approximation": Although I realize this is not the intent, the name "Single Derivative Order kernel" makes me think "first" derivative, rather than a single derivative of arbitrary order. I suggest a more explicit name like "$m$-order Derivative kernel".

4. The paper ends a bit abruptly. I think it would help the reader to end with a summary of the paper's key contributions and perhaps a discussion of the limitations and weaknesses of the proposed method.

**Questions:**

**Major**

1. A central idea of this paper is to estimate a square root of the target density function, in order to ensure non-negativity of the estimated density (after squaring). This is closely related to the proposal of [MFBR20], who propose a general framework for estimating non-negative quantities built on this idea. In the particular case of nonparametric density estimation, I think their proposal is very similar to the present paper's (namely, maximum likelihood with Sobolev norm regularization). They also propose various mechanisms to enforce the constraint that the density estimate integrates to $1$. How exactly does the present paper's proposal differ from that of [MFBR20], and what, if any, are the advantages?

2. Regarding the "positivity on the data points $x_i$" assumption in Appendix K: Does this assumption hold (for large $n$, with high probability) as $n \to \infty$?

**Minor**

1. Is there a motivation for explicitly enforcing the $L_2$ penalty $||f||_2$, as opposed to simply enforcing the Sobolev pseudo-norm penalty $||D f||_2$?

2. Equation (9): I think there is an extra square in the definition of this inner product (i.e., $\langle D^\kappa f, D^\kappa g \rangle^2$ should be $\langle D^\kappa f, D^\kappa g \rangle$).

3. Figure 1: What is "INER"? Should this be "RSR"?

4. Figure 2: What do the error bars indicate? Quantiles?

**References**

[MFBR20] Marteau-Ferey, U., Bach, F., & Rudi, A. (2020). Non-parametric models for non-negative functions. Advances in neural information processing systems, 33, 12816-12826.

---

> ### Author Response · Authors · 2023-11-16
>
> Thank you for the detailed feedback. We were glad to learn that the empirical results were found to be quite impressive.
>
> In the following we discuss all points raised in the review.
> In particular, we discuss the technical issue of density positivity, the relation to the work [MFBR20] (thanks for bringing it to our attention!), and the purpose of Section 5.  Please let us know whether some issues remain.
>
> In addition, we believe the reviewer may find the material in the global "General Reply" comment above to be of interest. There, we attempt to clarify the relation between various Sobolev space related kernels.
>
>
> First, we address the "positivity on the data points" question in the consistency result. This was raised in point 4 of the review and in the related Question 2.
>
> **positivity on the data points**:
>
> The reason we only look for solutions in the cone (104) with strictly positive $f(x_i)>0$ is that the gradient of the loss is not defined otherwise (Eq. (6) or (106)).  Note that the functions in the cone satisfy
> $f(x_i)>0$ simply because we defined them to satisfy this, thus, there is no contradiction here.
> In addition, we do want to evaluate the gradient of the *true* density $v$ at the sample points. We believe this was the source of the issue in the review, when $n\rightarrow \infty$.  However, this also does not constitute a problem.
> Indeed, note that we do not require a lower bound on $f(x_i)$, we just need to know the gradient is well defined, i.e., $f(x_i)\neq 0$.  To this end,
> for any *continuous*  density $v^2$, we have the following:
>
> Let $\{x_i\}$ be an infinite sequence of iid samples from $v^2$. Then $v(x_i)>0$ for all $i$ with probability 1. (since choosing $x$ with $f(x) =0$ is a 0 probability event).
>
> As an example, let $v^2$ be either a Gaussain, or a uniform density on a sphere. In both cases samples will have $v^2(x_i)>0$. The infimum of such values will  certainly be 0 for the Gaussian but there is no contradiction in that. Such points, approaching 0,  will not be optimal, and again, we only need the gradient to be well defined.
>
> Does this resolve the issue? We would be glad to discuss it further.
>
>
>
> Next we address the other major question, i.e., the relation to work [MFBR20].
>
> **relation to [MFBR20]**:
>
>
> At a very high level, the approach of [MFBR20] is indeed similar to ours, since they consider quadratic functions and regularisation with a version of an RKHS norm.
> The differences are both in scope and in details of the construction that enable computation.  From a computational viewpoint, they optimise in the space on non-negative matrices. This makes the problem convex without any need for positivity, but it requires $N^2$ parameters to encode such a matrix (their Theorem 1), and $O(N^3)$ computational cost per step (their top of page 6). Further, the optimisation is a constrained one, which is also significantly more difficult. It is unlikely their methods can work on something like ADBench benchmark, and indeed, they only test it on a 1 dim Gaussian with N=50.
>
> Second, while [MFBR20] is concerned with other applications as well, we
> investigate the density estimation problem much more closely.
> We prove consistency, which is completely orthogonal to all their results, we prove the method is different from KDE, and we do provide evaluation on a state of the art benchmark of datasets.
>
>
> Finally, they do treat the normalisation (their Prop 4, and paragraph below it).
> However, this is simply by a straight forward opening of the brackets.
> That is (in our case), if $f(x) = \sum_{i} \alpha_i k(x_i,x)$ then
> $\int f^2 (x)dx = \sum_{i,j} \alpha_i \alpha_j k(x_i,x)k(x_j,x)$.
>
> Thus, if one knows $\int k(x_i,x)k(x_j,x) dx$ for every $i,j$ one can compute  theoretically the normalisation constant. The issues with this are as follows: (a) This would involve summing $N^2$ floating point numbers, which would be inherently unstable for large N. (b) Computing
> $\int k(x_i,x)k(x_j,x) dx$ analytically is easy *only* for the Gaussian kernel. Even for exponential (Laplacian) kernel in $d>1$, we do not see any way to evaluate this analytically. Of course, for the SDO kernel, this also does not seem to be computable. Due to this reason, we have not pursued normalisation.
>
> It is worth noting however, that while the work of [MFBR20] advocates for using the computation above as a constraint, if one uses the Sobolev norms as we do,
> the integral is guaranteed to be finite, and it is thus sufficient to normalise only once, when the optimisation is finished.
>
>
>
> Continued in the comment below.

---

> > ### Author Response · Authors · 2023-11-16
> >
> > **Regarding point 1:**
> >
> > >I'm not convinced by this motivation, for a two reasons...
> >
> > >Thus, I'm not convinced that a new method will perform well in high dimensions unless it makes some more explicit assumptions on the density being estimated.
> >
> > Please note that density estimation from a finite sample is inherently about bias of the estimator, i.e., about assumptions. Without any assumptions, there is no way of going from a finite set of points to a density.
> > Thus all methods have a certain bias, and there no "best" method. This is well known in Anomaly Detection, for instance; see the ADBench paper.
> > In some methods the bias is clear, while in some, like neural networks, little is understood about it. We specifically were interested in a method with as simple bias as possible: the norm of a derivatives of one fixed order.
> >
> > We emphasize that empirically, at least, for Anomaly Detection, the estimator is very useful.
> >
> >
> > **Purpose of Section 5 (point 6):**
> >
> >
> > Our goal in Section 5 is to show that KDE and RSR can produce arbitrarily different results on the same data. Note that we do not claim here that either one is better for Anomaly Detection on the basis of this. As we mention above, it is unlikely that one can be better than the other universally.
> > However, we believe it is still important to understand why RSR is truly a new estimator, not equivalent to KDE. Proposition 3 and Figure 1  show this.
> > To show the estimators describe a different distributions, we consider the
> > ratio $f(x)/f(x')$ where $x,x'$ are points in first and second cluster.
> > If this ratio is larger, this means the estimator can change its value from one cluster to another faster.   (Here $f$ are the densities themselves, not the roots).
> >
> > In Fig 1b we have shown the log values of KDE and RSR estimators on both clusters. We have normalised them so that they have similar value on the first. We then see that log values of RSR are lower, meaning larger ratio, as expected from Proposition 3.
> >
> >
> > **Minor Questions:**
> >
> > 1) There are two reasons why having the term $\|f\|\_{L_2}^2$ in a norm such as (8) is desirable. First $\|Df\|\_{L_2}^2$ alone is only a semi-norm,
> > (it zeros out all constant functions), and thus it doesn't create a proper RKHS, causing issues with the representer theorem which is essential to obtain computable models. Second, in our setting this also conveniently guarantees that the function $f$ has a finite integral, so it can be renormalised to be a density.
> >
> > 2) yes, the square is a typo.
> >
> > 3) yes, its should be RSR.
> >
> > 4) yes, quantiles. The orange dot is the mean, the line in the middle is median. the large bar is 25%-75% quantiles.

---

> ### Comment · Reviewer_oZU9 · 2023-11-21
> **"positivity on the data points" condition**
>
> Thanks to the authors for their responses to my questions. The discussion of [MFBR20] was helpful to me; in particular, it seems that the optimization iterations of the proposed approach are $O(n^2)$, in contrast to the $O(n^3)$ iterations of [MFBR20], allowing the proposed method to scale to larger datasets.
>
> However, I didn't understand the authors' respose to my question about the "positivity on the data points" condition. Please let me know if I'm missing something here.
>
> > In addition, we do want to evaluate the gradient of the true density
>
> The parameters of the *true density* are fixed; what does it mean to evaluate "the gradient of the true density"? Doesn't (natural or standard) gradient descent actually require taking the gradient of $\alpha$ *at the current iterate*, not at the true value?
>
> To be explicit, let's consider the initial value $\alpha_0$. I believe computing the natural gradient (Eq. (6)) requires that the *estimated* density is positive on the data points, i.e., $f_0 := \min_{i = 1,...,N} \sum_{j = 1}^N \alpha_{0,j} k_{x_j}(x_i) > 0$. I agree that this is satisfied if $f_0$ lies in the positive cone. However, the paper claims that enforcing this cone constraint is computationally intractable and therefore suggests an alternative approach in practice, namely initializing each $\alpha_{0,j} \geq 0$. As noted in the paper, for a potentially negative kernel, such as the SDO kernel, $\sum_{i = 1}^N \alpha_{0,i} k_{x_i}(x)$ may still take negative values at some $x$. So it seems necessary to me to justify that, for $\alpha_{0,j} \geq 0$, $\min_{i = 1,...,N} \sum_{j = 1}^N \alpha_{0,j} k_{x_j}(x_i) > 0$.
>
> This seems unlikely to be true in general. In fact, it's easy to construct a counterexample when $N = 2$ (let $\alpha_{0,1} \gg \alpha_{0,2}$, and let $x_2$ be such that $k_{x_1}(x_2) < 0$). I think this creates a hole in the consistency argument. I was wondering, however, if there is a way to choose $\alpha$ such that this is true with probability approaching 1 for large $N$? If so, then this might not be a problem in practice, and showing this would solidify the consistency argument.
>
> EDIT: Another possibility that just occured to me is that the consistency guarantee is intended to be for the "ideal" algorithm, which exactly enforces the cone constraint. If this is the case, then I guess the guarantee stands, but there is a bit of a gap between the theory and the practice. Is this the intent?

---

> > ### Author Response · Authors · 2023-11-21
> >
> > Thank you for the response!
> >
> > >The parameters of the true density are fixed; what does it mean to evaluate "the gradient of the true density"? Doesn't (natural or standard) gradient descent actually require taking the gradient of $\alpha$ at the current iterate, not at the true value?
> >
> > We believe we might have caused some confusion with our previous comment, we will try to resolve this now.
> >
> > Lets consider the cone defined by Eq (104) .
> >
> > There are two scenarios here:  One is the scenario of the optimisation and the other is the scenario of the consistency algorithm.
> >
> >
> > **For the consistency result:**
> > Absolutely, as the reviewer mentions in the edit, in this part we do not specify how we have obtained the solution $u_N$ of the optimisation problem (104). By strong convexity of the log we know there is one in the cone, and it is unique, and the result is about this minimiser.
> >
> > What we meant by "evaluate the gradient of the true density" in the earlier comment is described in the end of this comment (as this not a crucial part of the discussion).
> >
> > **For the optimisation algorithm:**
> >
> > First note that if we use purely non-negative kernels of Sobolev type  (discussed in out global comment), such Gaussian and Laplacian, *and* we use the natural gradient, then initialising with  $\alpha_{0,i} > 0$ will ensure that we are in the cone,
> > and that the gradient iterations will leave us in the cone.
> >
> >
> > Next, for the SDO, this is somewhat more complicated. First, a random $\alpha_{0,i} > 0$ initialisation is not guaranteed any longer to place us inside the cone, and even if we are in the cone, the even natural gradient is not guaranteed to leave us there.
> >
> > Now two points arise:
> >
> > First, from the point of view if consistency, how do we actually find that minimal $u_N$ now? If we want to be fully strict, there indeed is no choice and we must use some other method to find the minimum. It will be computationally harder, but eventually it is a strictly convex objective with linear inequality constraints, it could be solved.
> >
> >
> > We believe this disconnect was the original issue.  While we emphasized in the text that SDO is not non-negative, we have not connected that fully with the consistency result, which is formulated in terms of the optimiser itself. For SDO kernel, the natural gradient can not guarantee finding the optimiser. **We thank the reviewer for this point and this discussion, and we will clarify this in the text.**
> >
> >
> > Second, at the same time, empirically, on real data, we do almost always get initialisation almost in the cone, and the natural gradient does mostly preserve the cone (and does so better than the regular gradient, as we verify in Section 6.3).  Thus natural gradient here is a
> > (well motivated) useful heuristic.
> >
> > Is this issue now clarified? Is this still an issue?
> >
> >
> >
> >
> >
> >
> > **What we meant by "evaluate the gradient of the true density"**
> > is simply that
> > our approach to the proof is to show that $\|\nabla L(v)\|\_{H}$ is small, where $L$ is the loss (which depends on $x_i$) and $v$ is the square root of the true density. (see the discussion around Eq. (106)-(108) ).
> >
> > Since the expression for $\|\nabla L(v)\|\_{H}$ involves
> > $v^{-1}(x_i)$,
> > we need these quantities to be non-zero. But this holds with probability one by the argument we outlined in the previous comment.
> >
> > But we understand now that these questions were not the original question.

---

> ### Comment · Reviewer_oZU9 · 2023-11-21
> **Clarifying my comment about the paper's motivation**
>
> Regarding my point about the paper's motivation
> > I'm not convinced by this motivation, for a two reasons...
>
> I fully agree with the authors' claim that
> > all methods have a certain bias, and there [is] no "best" method.
>
> In fact, I did not mean to question the motivation underlying the search for a new method (sorry it came across in this way!). Rather, I am commenting on the specific sentences
> > While there is recent work for low dimensional (one or two dimensional) data... there still are very few non-parametric methods applicable in higher dimensions.
>
> These sentences don't seem accurate or convincing to me, and I am suggesting that the authors re-write this paragraph to identify more specific strengths or motivations of their proposed method.

---

> > ### Author Response · Authors · 2023-11-21
> >
> > >These sentences don't seem accurate or convincing to me, and I am suggesting that the authors re-write this paragraph to identify more specific strengths or motivations of their proposed method.
> >
> > Our motivation for non parametric models is this:
> >
> > *Compared to parametric models, non parametric methods are often conceptually simpler, and the model bias (e.g., prior knowledge, type of smoothness) is explicit. This may allow better interpretability, and better regularization especially in smaller data regimes.*
> >
> > Would the reviewer agree with this? In particular, we certainly agree with the point from the original review that "[deep neural nets] they can fit quite complex densities".  But at least for now, it is not clear what exactly is their bias (i.e., how they generalise, how are they regularised) and they tend to require large sample sizes. We will add this clarification to the paper (as well as the [MFBR20] reference).
> >
> > Does this resolve the issue?

---

> > > ### Comment · Reviewer_oZU9 · 2023-11-22
> > >
> > > Thanks for the responses.
> > >
> > > I understand now that the consistency result assumes that the optimization is performed exactly, rather than using the heuristic initialization $\alpha \geq 0$. While this creates a bit of a gap between theory and practice, I am ok with this simplification, as long this is clarified in the paper. I think it would help significantly to put a formal statement of the consistency guarantee in the main paper rather than just the brief high-level description in Section 2.
> > >
> > > Regarding the motivations paragraph, I also agree, I think this is a more convincing motivation.
> > >
> > > Regarding the purpose of Section 5 (my original point 6.): I guess understand the intent of this section, but, the results here don't seem illuminating enough to devote 1.5 pages in the main paper. Although it shows that there is a difference between RSR and KDE in a particular example, it doesn't give a very clear or general idea of what this difference is. So, I think most of this could go in an appendix, with a few sentences in the main paper refering to this. This would create more space to describe the consistency result in detail.
> > >
> > > These responses are enough for me to raise my score, but I have to read through the other reviews and responses more carefully before deciding how much.

---

### Author Response · Authors · 2023-11-16

We thank the reviewers for their feedback and we appreciate the questions in the reviews.
We were also glad to learn that the reviewers found the paper clearly written (all three reviews), and the empirical results were found to be "impressive" (oZU9) and "good" (o1rt). Reviewers CL4E and o1rt further noted that the comparison the proposed method, RSR, with KDE is of interest.


In this comment we would like to emphasize a few aspects regarding the contributions of the paper, and also to discuss the specific SDO kernel and its relation to Matern kernels, addressing several questions in the reviews.

Detailed discussion of all other points raised in the reviews can be found in the individual review responses.


The contributions consisting of the introduction of the RSR objective,
the comparison of this objective to KDE, and the empirical evaluation of the final method were mentioned in the reviews.

However, the optimisation related contributions, which are crucial for the practical applicability of the method, were not discussed. We thus would like to emphasize them here:

The optimization-related contributions are: the  natural gradient with its positivity preserving properties (see the discussion following eq. (5) and also the discussion in our response in Review CL4E); the use of Fisher Divergence; and the sampling process in case of the SDO kernel, have not been discussed in the reviews. These are important steps in achieving the remarkable empirical performance of the method. Would the reviewers agree that these are new and significant contributions?


In addition, the consistency of the RSR estimator was not previously known beyond 1 dimension. Would the reviewers agree that this is a significant contribution?  While most of the details on this are in the supplementary material, the result is still part of the paper.



We now address a subject that was touched upon in a few reviews: the specific choice of the SDO kernel, which is also one of the new contributions of our paper. We discuss the motivation and the relation to other kernels. In addition to this discussion, we will provide additional experiments with other kernels in a few days, as discussed below.


Our motivation for introducing the SDO kernel and the associated norm was that it is the *simplest* kernel among the family of $L_2$ Sobolev type spaces. We believe its simplicity is important for interpretability purposes. This kernel was previously considered in the literature, [1],[2],[3],[4]. Analytic expressions in 1-d case were derived in [5]. Our interest here was in using the RSR estimator with this kernel in higher dimensions.

At the same time, as it was pointed out in review o1rt, and touched upon in review CL4E,
it may be of interest to consider other kernels with the RSR estimator,  and especially Matern kernels, related to the Sobolev space.
We agree with this remark, and are preforming experiments to evaluate RSR with the Laplacian and RBF kernels that are the opposite extremes of the Matern class (see below).  We will update with the results of these experiments in the coming days.
We note that *all the contributions in this paper are relevant for these kernels too*. Specifically, the RSR objective itself, the consistency proof, the optimisation via natural gradient, and the Fisher Divergence parameter tuning are all relevant.


We now discuss the relation between SDO, Laplacian, Gaussain and Matern kernels.


The SDO kernel with parameter $a$ is defined by the norm

$ \|f\|^2_{SDO}=\|f\|_{L_2}^2+ a \cdot \sum\_{|\kappa|_1 = m} \frac{m!}{\kappa!}\|{(D^{\kappa} f)}\|\_{L_2}^2.$

see our Equation (8). This involves only derivatives of order $m$ and we use $m \sim d/2$.


The RBF kernel with parameter $\sigma$ is given by (see [8],chapter 6)

$\|f\|\_{G}^2 = \|f\|_{L_2}^2+\sum\_{m=1}^{\infty}\frac{\sigma^{2m}}{m!2^m}\sum\_{|\kappa|_1 = m} \frac{m!}{\kappa!} \|{(D^{\kappa} f)}\|\_{L_2}^2.$

The Laplacian kernel, having the expression $k(x,y) = ({1/\sigma}^d) \cdot e^{-\|x-y\|/\sigma}$ (the norm here is the Euclidean norm) is given by

$\|f\|\_{L} = \|f\|\_{L_2}^2 + \sum\_{i=1}^{(d+1)/2} \sigma^{2m} C_{d,m} \sum\_{|\kappa|_1 = m}   \frac{m!}{\kappa!} \|{(D^{\kappa} f)}\|\_{L_2}^2.$

for odd $d$. Here $C_{d,m}$ is a constant independent of $\sigma$; see [7], Proposition 7. The Laplacian kernel is also known as the Exponential kernel.

Continued in the comment below.

---

> ### Author Response · Authors · 2023-11-16
>
> Thus, the Laplacian requires $d/2$ derivatives, a minimal requirement for the Sobolev RKHS, and the Gaussain requires infinite smoothness. The Matern kernels have (quite recently) been shown to interpolate between these extremes, requiring $\nu > d/2$ derivatives. The Laplacian and Gaussian are  special instances of this class; see, for instance, Example 5.7 in [6].
>
>
> The relation between these kernels is now clear. The Laplacian, Gaussian and Matern kernels enjoy closed analytic forms, but this comes at the expense of a quite complex weighting of different orders of derivatives,
> and the bandwidth parameters $\sigma$ affecting all of the orders, but with an exponential scaling. Imagine trying to explain this for instance to a medical doctor, in a context of interpretability discussion of your model. On the other hand, SDO as we use here is similar to the Laplace in requiring only a $d/2$-th smoothness, but we measure only the size of that derivative.
>
> Finally, we would like to mention that for Matern kernels, the positivity preserving properties of the natural gradient apply *exactly* (see the discussion following Eq. (5)) rather than just approximately, since these are non-negative kernels.
>
>
> We hope this clarifies the contribution. The empirical evaluation of RSR with Laplace and Gaussian kernels will be included in the few coming days.
> We would be glad to answer any further questions.
>
>
>
>
>
> [1] Cox, Multivariate smoothing spline functions, SIAM Journal on Numerical Analysis, 1984
>
> [2] Delecroix, Simioni Thomas-Agnan, Functional estimation under shape constraints,  Journal of Nonparametric Statistics, 1996
>
> [3] Novak, Ullrich, Wo{\'z}niakowski,Zhang, Reproducing kernels of Sobolev spaces on R^d and applications to embedding constants and tractability,
> Analysis and Applications, 2018
>
> [4] Berlinet,  Thomas-Agnan, Reproducing kernel Hilbert spaces in probability and statistics,Springer Science, 2011
>
> [5] Thomas-Agnan, Computing a family of reproducing kernels for statistical applications, Numerical Algorithms, 1996
>
> [6] Fasshauer, Ye, Reproducing kernels of generalized Sobolev spaces via a Green function approach with distributional operators, Numerische Mathematik, 2011
>
> [7] Rakhlin, Zhai, Consistency of interpolation with Laplace kernels is a high-dimensional phenomenon, Conference on Learning Theory, 2019
>
> [8] Rasmussen, Williams, Gaussian processes for machine learning, 2006, Springer

---

### Author Response · Authors · 2023-11-20
**paper revision**

Dear Reviewers,


We have just completed the experiments evaluating RSR on additional kernels, and have updated the paper.The experiment consisted of evaluating RSR on the Laplacian and Gaussain kernels, which are extremes of the Matern family, as discussed in the previous global comment. The results are in Supplementary Material Section M.


We have found that all kernels (SDO, Laplacian, Gaussain) perform similarly of the ADBench task, and thus RSR is very competivtive compared to other anomaly detection methods,  for all the kernels.Recall that as we mentioned in the previous comment, the SDO is conceptually (although not computationally) the simplest Sobolev type kernel. At the same time, we have also discussed that all the theoretical and methodological contributions of this paper for the RSR estimator, apply equally to all
such kernels, and in particular to the Laplacian, Gaussain, and all other Matern kernels.



In addition to the above evaluation, we have implemented several of the stylistic/typographical suggestions from the reviews. We will implement additional ones in the final version of the paper. In particular, as suggested in the reviews oZU9 and CL4E, we will move a more significant part of the discussion of the consistency results from the Supplementary to the main paper. Due to space constraint this will require a considerable restructuring of the material.

---

### Meta-Review · Area_Chair_uEyM · 2023-12-05

**Metareview:**

This paper proposes a method of kernel density estimation using the Sobolev norm as a regularizer. The methodology is simple, but the analysis and the methodology are nicely and carefully designed. Reflecting the long accumulation of kernel density estimation, the authors' arguments and careful and well thought-out. However, this paper did not receive positive support from the reviewers. One reason is that kernel estimation using such an approach is too commonplace, and the key idea of this paper did not seem novel. Some reviewers pointed out there are similar approaches, and this paper's novelty is limited (of course, there is a possibility that it is technically very nontrivial, but at least this paper is not written in a way that would persuade us of that). This paper should better organize and differentiate existing research and present key ideas more clearly.

**Justification For Why Not Higher Score:**

The weaknesses of this paper are clearly identified and there is consensus among the reviewers that there is room for improvement.

**Justification For Why Not Lower Score:**

N/A

---

### Decision · Program_Chairs · 2024-01-16

Reject